# CAN MONOCULAR SINGLE-DEPTH FOUNDATION MODELS GENERATE MULTI-DEPTH HYPOTHESES?

## ABSTRACT

Monocular depth foundation models underpin modern 3D perception, yet they are mainly trained under a restrictive paradigm that predicts a single deterministic depth value per pixel. This formulation assumes that every image ray intersects only one surface, an assumption that fails in transparent or multi-layer scenes. This raises our central question: can models built for single-depth prediction generate multi-depth hypotheses? We find that they can. Beneath their deterministic outputs lies latent multi-layer structure, a hidden geometry obscured by the training objective rather than absent from the model itself. To uncover it, we introduce Laplacian Visual Prompting (LVP), a lightweight input-space perturbation that elicits complementary depth hypotheses from a frozen model without retraining. On our new MD-3k benchmark of ambiguous scenes, LVP consistently decouples foreground and background layers, showing that a single off-the-shelf depth foundation model can be reprogrammed into a multi-hypothesis estimator. These results reveal that the geometric capacity of depth foundation models is richer than their standard outputs suggest, and open a new path toward probing and harnessing hidden representations for more complete 3D understanding under ambiguity.

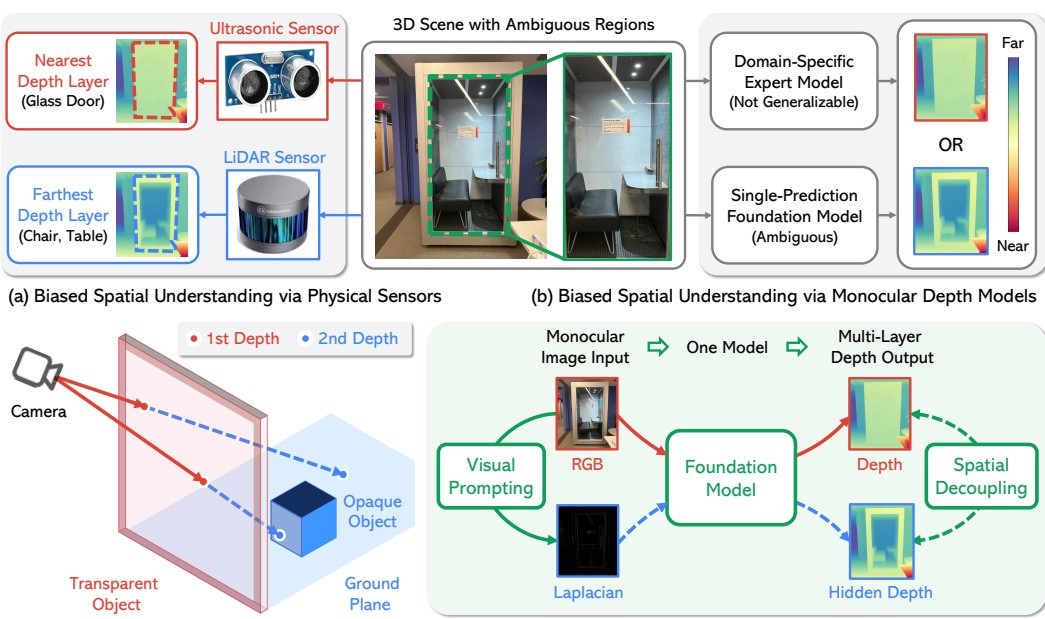

Figure 1: **Is There More Than One Depth?** While ambiguous scenes (**c**) contain multiple depth layers, existing systems produce a single, biased result. (**a**) Physical sensors give contradictory, single-layer outputs (near vs. far). (**b**) Monocular foundation models, trained for one output, inherit this bias. (**d**) We show how to unlock the hidden, multi-layer understanding already present in these models. By probing a single, frozen model with an RGB input and our novel **Laplacian Visual Prompt (LVP)**, we elicit two distinct, complementary depth hypotheses, all without retraining.

# 1 INTRODUCTION

For any given pixel in an image, what is its depth? For decades, our algorithms have been built on the premise of a single, unambiguous answer. We build massive monocular depth foundation models (Ranftl et al., 2022; Yang et al., 2024a;b) and train them on vast datasets, all to produce one depth value per pixel. However, this core assumption is fundamentally at odds with the real world (Wen et al., 2025). What happens when a line of sight intersects not one but multiple surfaces, as with a simple pane of glass (Fig. 1c) in transparent scenes? **Is there more than one depth?**

The single-depth prediction paradigm is inherited from the real-world depth maps collected by physical sensors, which are themselves **biased**. An ultrasonic sensor perceives only the closest surface, while LiDAR can see through it, capturing only the farthest objects (Fig. 1a). It is no surprise that our computational large-scale models inherit this limitation. When faced with a transparent surface, a powerful pre-trained model is forced to give **a single, incomplete answer**, producing a biased depth map that captures only one aspect of the scene's geometry (Fig. 1b). This is typically seen as a model failure. **We argue that it is a failure of the *question* we are asking for the model.**

What if, instead of asking for the depth, we could probe the monocular depth foundation model for its richer, latent spatial understanding of a scene? Surprisingly, we find that **foundation models supervised with single-layer depth outputs already contain this multilayered knowledge**, dormant within their weights. This finding parallels the recent discovery of other **emergent capabilities** in visual foundation models, which can show surprising zero-shot abilities for correspondence (Tang et al., 2023b; An et al., 2025), motion (Chen et al., 2025a), and other tasks (Wiedemer et al., 2025). We show that without any fine-tuning or architectural changes, we can elicit multiple, distinct depth hypotheses from a single, off-the-shelf depth foundation model (Fig. 1d). This means that the model knows about both the transparency and the opaque object behind it.

Our probe is almost trivially simple: a Laplacian-transformed image. As a cornerstone of classical computer vision, the Laplacian operator excels at edge detection by isolating high-frequency components in an image. We leverage this fundamental property not for edge detection itself but as a visual question. As shown in Fig. 2, by feeding the model this high-frequency version of the image, we are essentially asking it to ignore smooth, low-frequency surfaces (*e.g.*, a glass) and report on the scene's detailed, high-frequency structures (like the background behind it). In response, the same frozen model produces a second complementary depth map that it had previously ignored. This training-free process, which we term **Laplacian Vision Prompting (LVP)**, is not only simple and fast, but also entirely avoids any learnable parameters, allowing us to decouple a scene's layers using the model's own latent knowledge.

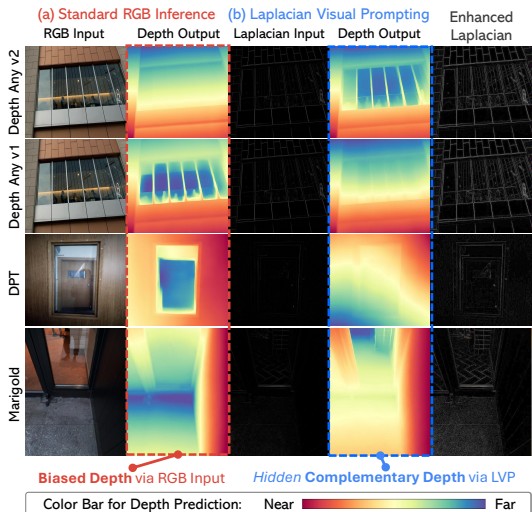

Figure 2: **Unlocking Hidden Depth Layer in Diverse Depth Foundation Models.** A standard RGB input compels a pre-trained monocular depth foundation model to output its default, biased depth (**column 2**). Our Laplacian visual prompt asks the same frozen model a different question, revealing a second, **complementary 'hidden depth'** map in ambiguous transparent regions (**column 4**).

To facilitate this investigation, we introduce **MD-3k**, a new real-world benchmark for studying these multi-layer phenomena. We use it to systematically characterize the biases of various foundation models before showing how LVP can elicit their hidden spatial understanding. Our findings suggest that the path to more robust 3D perception may not always require bigger models, but cleverer ways of interrogating the vast knowledge already latent within the foundation models.

In summary, our contributions are: **1**) We rethink the monocular single-depth estimation paradigm, framing the problem as one of probing the latent, multi-hypothesis understanding of foundation models. **2**) We introduce MD-3k, a real-world benchmark to evaluate and diagnose model biases in multi-layer spatial perception. **3**) We propose Laplacian Vision Prompting (LVP), a simple, training-

free method to elicit a second, complementary depth map from a single frozen foundation model. **4**) We show that LVP can elicit two-layer depth across diverse models under ambiguous transparent scenes, enabling more flexible geometry-conditioned image generation and video inference.

## 2 RELATED WORK

**Monocular depth estimation foundation models.** Our methodology leverages the capabilities of modern, large-scale monocular depth estimation (MDE) foundation models. The field has progressed from domain-specific models trained on datasets like KITTI (Geiger et al., 2013) and NYUv2 (Silberman et al., 2012; Eigen et al., 2014; Fu et al., 2018; Bhat et al., 2021), to general-purpose, domain-agnostic systems. These models achieve remarkable zero-shot performance by using diverse mixed training data (Ranftl et al., 2022; 2021; Birkl et al., 2023; Yin et al., 2023), using generative priors from large diffusion models (Rombach et al., 2022; Ke et al., 2024; Gui et al., 2024; Fu et al., 2024), or training on massive-scale pseudo-labeled datasets (Yang et al., 2024a;b). Despite their sophistication, most of these models are architecturally and functionally *limited to a single-depth prediction paradigm*, which outputs one depth value per pixel. This design is the root cause of the layer bias we identify in ambiguous scenes. Our work therefore poses a central question: *Do these models implicitly learn multi-layer representations that can be accessed without re-training?*

**Depth estimation for ambiguous scenes with transparency.** Reliable 3D perception in the presence of transparent materials is a long-standing challenge, as standard monocular cues become ambiguous. Early research predominantly framed the task as a *single-layer completion problem*, aiming to recover a single coherent depth map by inpainting missing sensor data (Sajjan et al., 2020; Zhu et al., 2021) or by having a model learn to regress the depth of the background scene (Chen et al., 2023b; Fang et al., 2022). While effective for specific applications, this paradigm fundamentally collapses the valid multi-layer geometry into a single, biased estimate. Acknowledging this limitation, more recent work has shifted towards *multi-layer inference*, aided by new benchmarks with multi-layer annotations (Wen et al., 2025). These methods often employ specialized architectures (Zhang et al., 2024a) to predict multiple depth layers. However, they typically require custom model designs and complex, supervised training objectives. In contrast, our work investigates whether **multi-layer depth can be elicited from existing, general-purpose models without any retraining**. This distinguishes our approach from generative sampling; for instance, while diffusion models (Ji et al., 2023) could theoretically sample multiple depth hypotheses via different noise seeds—an avenue that remains unexplored—such a process would be non-deterministic and computationally expensive. LVP, in contrast, is deterministic and highly efficient, requiring only an additional forward pass.

**Visual prompting for model adaptation.** The mechanism we use to access the latent knowledge is Visual Prompting (VP) (Bahng et al., 2022; Bai et al., 2024), a paradigm for adapting pre-trained models inspired by prompting in NLP (Brown et al., 2020). VP has been successfully applied to vision-language models (Singha et al., 2023; Wasim et al., 2023; Khattak et al., 2023; Wang et al., 2024a) and for tasks such as cross-domain transfer (Chen et al., 2021; Neekhara et al., 2022) and improving adversarial robustness (Chen et al., 2023a). While many techniques focus on adding or tuning learnable parameters, a distinct sub-area is *input-space prompting*, where the input signal itself is modified to steer model behavior (Tsai et al., 2020). However, *the potential of VP for interrogating and decoupling low-level 3D geometric representations remains largely unexplored*. Our work falls into this category. We introduce Laplacian Visual Prompting as a simple, non-learned, input-space prompt that can compel a frozen foundation model to produce a complementary hypothesis.

## 3 METHOD

We propose a framework for resolving ambiguity in monocular depth estimation under transparency. Our method builds on three key components: **1**) a probabilistic formulation of multi-layer depth estimation (ML-MDE), **2**) the `MD-3k` benchmark and metrics to quantify bias, and **3**) Laplacian Visual Prompting (LVP), which deterministically elicits complementary hypotheses from frozen depth models. We also provide a theoretical analysis showing that multi-hypothesis predictions and input prompts reduce risk and provably improve multi-layer recovery.

### 3.1 PROBLEM FORMULATION: DEPTH AS A MULTIMODAL POSTERIOR

Let $\mathcal{I} \in \mathbb{R}^{H \times W \times 3}$ be an RGB image. The goal of monocular depth estimation is to infer a corresponding depth map $\mathcal{D} \in \mathbb{R}^{H \times W}$. From a Bayesian perspective, this can be framed as finding the Maximum a Posteriori (MAP) estimate of the depth map given the image:

$$\hat{\mathcal{D}}_{\mathrm{MAP}} = \arg \max_{\mathcal{D}} p(\mathcal{D}|\mathcal{I}). \tag{1}$$

Standard deep learning models for MDE, denoted $f_\theta$, are trained on large datasets of $(\mathcal{I}, \mathcal{D})$ pairs to produce a deterministic estimate, $f_\theta(\mathcal{I}) = \hat{\mathcal{D}}$, which implicitly approximates this MAP solution. This paradigm is effective when the posterior distribution $p(\mathcal{D}|\mathcal{I})$ is sharply ***unimodal***, as is the case for typical non-ambiguous opaque scenes.

However, in scenes containing transparent surfaces, the mapping from image to 3D structure is no longer one-to-one. A single line of sight can contain multiple surfaces, inducing a ***multimodal posterior distribution*** over depth maps. For a typical scene with a transparent foreground and an opaque background, this distribution may exhibit at least two distinct modes, corresponding to the depth of the transparent surface, $\mathcal{D}_1$, and the depth of the background, $\mathcal{D}_2$. A standard model $f_\theta$, architecturally constrained to produce a single output, is forced to select only one of these modes. The specific mode it consistently chooses—whether by dataset statistics or architectural priors—constitutes its ***inherent depth prediction layer bias***.

This leads to our formulation of **Multi-Layer Monocular Depth Estimation (ML-MDE)**. The objective is not to find a single MAP estimate, but to identify the set of primary modes $\{\mathcal{D}_1, \ldots, \mathcal{D}_K\}$ of the posterior. For the common two-layer case (*e.g.*, transparent foreground with opaque background), we seek to recover the ordered pair of depth maps that jointly maximize the posterior:

$$(\hat{\mathcal{D}}_1, \hat{\mathcal{D}}_2) = \arg \max_{(\mathcal{D}_1, \mathcal{D}_2)} p(\mathcal{D}_1, \mathcal{D}_2|\mathcal{I}), \quad \text{s.t.} \quad \forall_{u,v}\, \mathcal{D}_1(u,v) \le \mathcal{D}_2(u,v). \tag{2}$$

### 3.2 MD-3K BENCHMARK FOR EVALUATING MULTI-LAYER SPATIAL RELATIONSHIPS

Directly modeling the joint posterior in Eq. 2 is challenging and would require specialized architectures and multi-layer ground-truth data for training. Our work instead investigates a novel approach: ***can we recover the primary modes*** $(\hat{\mathcal{D}}_1, \hat{\mathcal{D}}_2)$ ***by strategically querying a standard, pre-trained model*** $f_\theta$ ***just like the way we prompt a LLM***? We hypothesize that a model's default prediction $f_\theta(\mathcal{I})$ approximates one mode, and a carefully prompted prediction can reveal another. To validate this, we created the **M**ulti-layer **D**epth **3k** (MD-3k) benchmark to provide a rigorous testbed for ML-MDE.

**Benchmark construction.** MD-3k consists of 3,161 Real-World RGB images sourced from the GDD dataset (Mei et al., 2020), selected for depth ambiguity, such as transparency. MD-3k is not filtered to only include ideal scenes (e.g., smooth glass, high-frequency background), making it a robust test of generalization. Following previous spatial relationship benchmarks for non-ambiguous scenes (*e.g.*, DIW (Chen et al., 2016) and DA-2k (Yang et al., 2024b)), we randomly sample sparse point pairs within the ambiguous region for each image. Expert annotators assigned pairwise depth order labels to points both on and behind transparent surfaces, generating two annotation layers. Annotation accuracy was rigorously validated through multi-round expert review. As shown in Fig. 3, each sample includes an RGB image, segmentation masks of ambiguous regions, and two types of spatial relationship labels (*near*

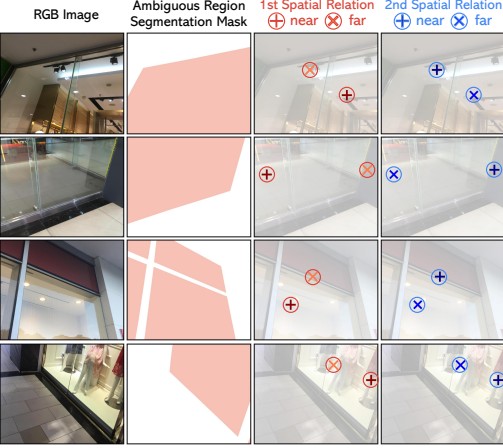

Figure 3: **MD-3k benchmark for evaluating multi-layer spatial relationships.**

and *far*) for point pairs representing multi-layer depths. The top three rows depict ***reverse*** relationships between two layers, while the bottom row shows ***same*** relationship between layers.

**Benchmark statistics.** The full `MD-3k` dataset, referred to as *overall*, is divided into two subsets with different multi-layer spatial relationships for fine-grained analysis: **1)** *Same* subset (1,783 point pairs): Consistent multi-layer relative depth ordering for each point pair; **2)** *Reverse* subset (1,378 point pairs): Reversed multi-layer relative depth ordering for each point pair. These subsets facilitate the evaluation of models under varying conditions of multi-layer spatial relationships. Fig. 4 summarizes statistics of ambiguous regions in the

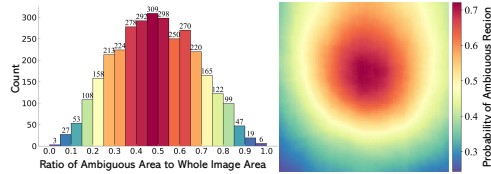

Figure 4: **Statistics of ambiguous regions in the `MD-3k` benchmark.** Ratio of ambiguous regions to the whole image (**Left**) and spatial distribution of ambiguity regions (**Right**)

`MD-3k` benchmark. The left panel shows a histogram of the ambiguous-to-total area ratio per sample, capturing diverse ambiguity levels from minimal to near-total scene ambiguity. The right panel's heatmap indicates a balanced spatial distribution with a slight center bias, resembling a Gaussian pattern that reflects natural scene compositions while minimizing regional biases.

### 3.3 EVALUATION METRICS

We introduce three metrics to separate model accuracy from bias.

**Spatial Relationship Accuracy (SRA).** For a given layer $i \in \{1, 2\}$, SRA$(i)$ measures the percentage of point pairs where the predicted relative depth is correct:

$$\text{SRA}(i) = \frac{1}{|\mathcal{P}|} \sum_{(P_1, P_2) \in \mathcal{P}} \mathbb{I}\Big(\text{sign}(\hat{d}_1^{(i)} - \hat{d}_2^{(i)}) = \text{sign}(d_1^{(i)} - d_2^{(i)})\Big), \quad (3)$$

**Depth Layer Preference ($\alpha$).** The key to our method is diagnosing a model's mode-selection bias. We define Depth Layer Preference $\alpha(f_\theta)$ as the difference in SRA between the two layers:

$$\alpha(f_\theta) = \text{SRA}(2) - \text{SRA}(1). \quad (4)$$

This metric provides a direct, empirical signal of a model's tendency: $\alpha > 0$ indicates a preference for the background mode, while $\alpha < 0$ indicates a preference for the foreground (transparent surface) mode. This diagnosed preference is not a flaw, but a predictable behavior to be exploited.

**Multi-Layer SRA (ML-SRA).** To measure overall success, ML-SRA calculates the fraction of point pairs where the relative depth is correctly predicted for *both* layers simultaneously:

$$\text{ML-SRA} = \frac{1}{|\mathcal{P}|} \sum_{(P_1, P_2) \in \mathcal{P}} \mathbb{I}\left(\bigwedge_{k=1}^{2} \text{sign}(\hat{d}_1^{(k)} - \hat{d}_2^{(k)}) = \text{sign}(d_1^{(k)} - d_2^{(k)})\right). \quad (5)$$

### 3.4 LAPLACIAN VISUAL PROMPTING (LVP)

Frozen monocular depth foundation models can inherit inductive biases from training data that collapse multimodal posteriors into a single prediction. To unlock hidden depth hypotheses, we propose Laplacian Visual Prompting (LVP), a deterministic input transformation that exploits these biases. By altering input frequency statistics, LVP redirects the model toward an alternative posterior mode, yielding multi-layer depth predictions without any retraining and finetuning.

**Laplacian operator as a high-pass prompt.** The Laplacian operator,

$$\Delta = \frac{\partial^2}{\partial x^2} + \frac{\partial^2}{\partial y^2}, \quad (6)$$

acts as an isotropic high-pass filter. Applied to $\mathcal{I}$, it suppresses smooth, low-frequency transparency cues from foreground glass while amplifying sharp, high-frequency

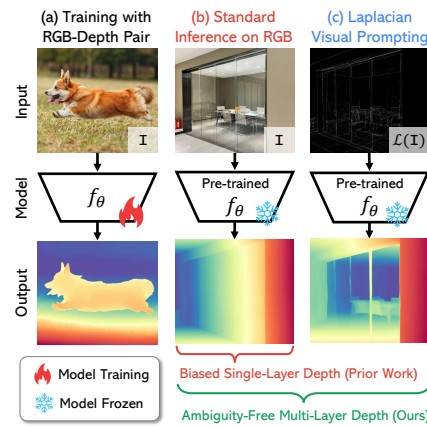

Figure 5: **The LVP Method.** (**a**) A frozen depth model $f_\theta$. (**b**) A standard RGB input yields a biased single-layer depth estimate. (**c**) An LVP-transformed input elicits an alternative depth.

background edges. This isotropy makes the operator robust to object orientation. Thus, the LVP-transformed image $\mathcal{L}(\mathcal{I})$ functions as a prompt that encourages the frozen depth foundation model to predict a complementary hypothesis. In discrete form, the Laplacian operator is approximated via a second-order finite difference scheme. Throughout this paper, our default LVP implementation applies a standard $3 \times 3$ Laplacian kernel to each channel of the RGB input image, defined as:

$$\mathcal{M}_\mathcal{L} = \begin{bmatrix} 0 & 1 & 0 \\ 1 & -4 & 1 \\ 0 & 1 & 0 \end{bmatrix}. \tag{7}$$

**Two-layer inference via bias exploitation.** Let $f_\theta$ be a monocular depth model pre-trained on large-scale image and depth map pairs (see Fig. 5a). Presented with an ambiguous RGB image $\mathcal{I}$, the model produces a biased single-layer estimate: $\mathcal{D}_{\mathrm{RGB}} = f_\theta(\mathcal{I})$ (see Fig. 5b). Feeding the LVP-transformed input yields a complementary prediction: $\mathcal{D}_{\mathrm{LVP}} = f_\theta(\mathcal{L}(\mathcal{I}))$. Together, $(\mathcal{D}_{\mathrm{RGB}}, \mathcal{D}_{\mathrm{LVP}})$ represent two hypotheses corresponding to distinct posterior modes (see Fig. 5c). This procedure converts a biased single-layer model into a deterministic two-layer predictor that directly addresses the ML-MDE problem, without any model optimization or finetuning. The ordering of the two derived depth maps can be determined in two ways: either by directly comparing their depth values on a pixel-wise basis or by using the model's pre-diagnosed depth layer preference.

### 3.5 THEORETICAL ANALYSIS

We provide a theoretical justification for why Laplacian Visual Prompting (LVP) can improve the probability of accurate multi-layer scene recovery. The core argument is that LVP acts as a principled operator to isolate distinct modes of the multimodal depth posterior that arises in ambiguous scenes.

**Proposition 1** (Mode Isolation via Prompting). *Let $p(\mathcal{D} \mid \mathcal{I})$ be a multimodal posterior distribution over depth maps $\mathcal{D}$ given an image $\mathcal{I}$, with primary modes at the foreground depth $\mathcal{D}_1$ and background depth $\mathcal{D}_2$. Let a model $f_\theta$ approximate the MAP estimate, such that its standard prediction $f_\theta(\mathcal{I}) \approx \mathcal{D}_1$. An ideal complementary prompt is an operator $T$ that transforms the input to $T(\mathcal{I})$ such that the posterior is re-weighted, causing the model's prediction to shift to the second mode:*

$$f_\theta(T(\mathcal{I})) \approx \arg\max_{\mathcal{D}} p(\mathcal{D} \mid T(\mathcal{I})) \approx \mathcal{D}_2 \tag{8}$$

**Justification of LVP as a mode-switching prompt.** We hypothesize LVP acts not as a fixed layer-finder, but as a **mode-switching operator** for some models. A model $f_\theta$ has a default, primary mode (e.g., $\mathcal{D}_1$) when given a standard input $\mathcal{I}$. LVP, as a strong OOD perturbation $\mathcal{L}(\mathcal{I})$, alters the model's internal feature representation (as shown in Fig. 7 in the experiment section)) just enough to *knock* it from its default mode into the next most stable posterior mode it has learned (e.g., $\mathcal{D}_2$). This *mode-switching* hypothesis explains why the effect of LVP is *complementary* to the model's default bias: models that default to $\mathcal{D}_1$ (like DAv2) are prompted to find $\mathcal{D}_2$, while models that default to $\mathcal{D}_2$ (like DPT) are prompted to find $\mathcal{D}_1$ (see Fig. 6 in the experiment section).

This mechanism is supported by our observation that models were trained on high-frequency signals (edges, textures) present in natural RGB images; LVP simply isolates this signal. The fact that some model variants *generalizes* and produces a *alternative* geometric structure, rather than failing catastrophically on this OOD input, is the core emergent capability we are highlighting.

From this principle, the following benefits are direct consequences:

**Corollary 1** (Multi-Hypothesis Risk Reduction). *Let $H(\mathcal{I}) = \{f_\theta(\mathcal{I}), f_\theta(\mathcal{L}(\mathcal{I}))\}$ be the two hypotheses approximating distinct modes $\{\mathcal{D}_1, \mathcal{D}_2\}$. The multi-hypothesis risk, $R_{\mathrm{multi}} = \mathbb{E}\left[\min_{k \in \{1,2\}} \ell(\hat{\mathcal{D}}_k, \mathcal{D})\right]$, is strictly less than the single-hypothesis risk, $R_{\mathrm{single}} = \mathbb{E}\left[\ell(\hat{\mathcal{D}}_1, \mathcal{D})\right]$, if the posterior is multimodal and LVP successfully isolates a valid complementary mode.*

**Corollary 2** (Improved Multi-Layer Accuracy). *Let $A$ be the event of a correct layer-1 relation from RGB input (with probability $p_A$) and $B$ be the event of a correct layer-2 relation from LVP input (with probability $p_B$). The probability of recovering both layers correctly is bounded by the Fréchet inequality: $\mathbb{P}(A \cap B) \geq \max\{0, p_A + p_B - 1\}$. If each prompt successfully targets its respective layer s.t. $p_A > 0.5$ and $p_B > 0.5$, the probability of correct multi-layer recovery is strictly positive.*

## 4 EXPERIMENTS

**Research questions.** Our experiments are designed to answer five key questions. **1**) What biases do existing depth models exhibit in ambiguous scenes? **2**) Can LVP exploit these biases for accurate multi-layer depth? **3**) How does model scale influence performance? **4**) What are the practical benefits of multi-layer depth maps? **5**) How well does LVP perform relative to alternative prompts?

**Experimental setup.** We evaluate a diverse set of pre-trained monocular depth foundation models, including the Depth Anything series (DAv1/2-S,B,L) with ViT backbones (Yang et al., 2024a;b), including general-purpose and domain-specific (Indoor/Outdoor) variants. We also test other prominent models, including discriminative architectures like DPT (Ranftl et al., 2021) and ZoeDepth (Bhat et al., 2023), generative models like Marigold (Ke et al., 2024) and GeoWizard (Fu et al., 2024), and metric depth models like Depth Pro (Bochkovskii et al., 2024), Metric3D-v2-L (Hu et al., 2024), and UniK3D-L (Piccinelli et al., 2025a). All experiments are performed in a training-free manner.

### 4.1 DIAGNOSING DEPTH LAYER BIAS

We begin by using our MD-3k benchmark and the Depth Layer Preference ($\alpha$) metric to diagnose the inherent biases of single-layer models.

**Models exhibit heterogeneous biases under the standard RGB image input.** As shown in Fig. 6 (RGB columns), pre-trained models exhibit strong but inconsistent depth layer preference biases. Some models, such as the series of DAv2 and DAv2-I mmodels, are strongly biased towards the first layer ($\alpha < 0$), while the outdoor-finetuned DAv2-O, DAv1, and other models are biased towards the second layer. This demonstrates that a model's final prediction in an ambiguous scene is a product of its architecture and training data, rather than a consistent interpretation of 3D space.

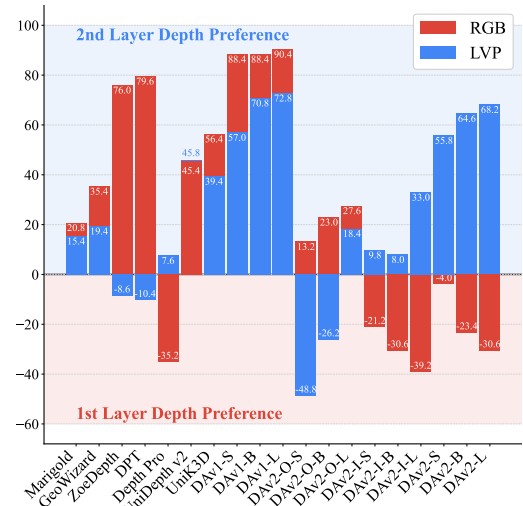

Figure 6: **Depth Layer Preference $\alpha$ [%] under RGB and LVP on *Reverse* subset of MD-3k.**

**LVP modulates layer preference.** As shown in Fig. 6, LVP can *reverse* depth prediction layer biases for some models. For example, in DAv2 and Depth Pro, the preference shifts from the first layer (RGB) to the second (LVP), while in DPT, ZoeDepth, and DAv2-O-S/B, it flips from the second layer (RGB) to the first (LVP). This result supports our *mode-switching* hypothesis: LVP is not a simple *background-finder* but an operator that generates *a complementary* mode to the model's default bias under standard RGB image inputs.

### 4.2 EVALUATING MULTI-LAYER DEPTH EMPOWERED VIA LVP

Having shown that LVP can generate complementary depth hypotheses, we now evaluate the accuracy of the final multi-layer predictions using the ML-SRA metric (see Table 1).

**LVP uncovers multi-layer spatial understanding.** As shown in Table 1, our DAv2-L model achieves a 75.5% *Overall* score. To contextualize this result, we compare against a **ideal** strong heuristic single-depth prediction model baseline that simply copies the primary (RGB) depth to the secondary depth. This baseline, by definition, achieves 100% on the *Same* subset (1,783 pairs) and 0% on the *Reverse* subset (1,378 pairs), yielding a weighted *Overall* score of 56.4%. Our model's performance thus represents a **+19.1%** absolute improvement, demonstrating that LVP elicits distinct and meaningful geometric information. This result indicates that some depth foundation models, *despite single-layer training, implicitly encode multi-layer representations* that LVP can decouple.

**Subset performance reveals ambiguity challenge.** On the simpler *Same* subset (consistent depth relationship across layers), models perform exceptionally well (all > 83% in ML-SRA). However, on the highly ambiguous and challenging *Reverse* subset, the best model DAv2-L only achieves 52.2%.

Table 1: **Multi-Layer Spatial Relationship Accuracy (ML-SRA) of LVP-elicited multi-layer depth on `MD-3k`.**

| Setting | Overall | Reverse | Same |
|---|---|---|---|
| *Single-Depth*[†] | 56.4 [†] | 0.0 | 100.0 [†] |
| Marigold | 57.4 | 15.3 | 89.8 |
| GeoWizard | 59.5 | 17.6 | 91.9 |
| ZoeDepth | 68.8 | 45.4 | 86.8 |
| DPT | 70.2 | 46.4 | 88.7 |
| Depth Pro | 66.3 | 31.1 | 93.5 |
| UniDepth-v2-L | 61.3 | 13.6 | 93.7 |
| UniK3D-L | 58.9 | 19.2 | **93.9** |
| DAv1-S | 57.9 | 17.7 | 89.0 |
| DAv1-B | 56.6 | 11.4 | 91.5 |
| DAv1-L | 57.1 | 10.9 | 92.8 |
| DAv2-O-S | 63.0 | 36.6 | 83.5 |
| DAv2-O-B | 62.7 | 32.9 | 85.6 |
| DAv2-O-L | 60.4 | 17.6 | 93.4 |
| DAv2-I-S | 60.9 | 27.7 | 86.5 |
| DAv2-I-B | 63.7 | 28.1 | 91.2 |
| DAv2-I-L | 71.1 | 42.5 | 93.2 |
| DAv2-S | 67.2 | 36.9 | 90.7 |
| DAv2-B | 73.3 | 48.2 | 92.7 |
| DAv2-L | **75.5** | **52.2** | 93.6 |

[†] **Ideal upper-bound performance** for a **single-layer** depth model that only produces one depth prediction.

Table 2: **Comparison of ML-SRA using LVP alone vs. using extra strong semantic priors on DAv2-L baseline.**

| Method | Semantics | Overall | Reverse | Same |
|---|---|---|---|---|
| **LVP (Default, Ours)** | No | 75.5 | 52.2 | 93.6 |
| + Predicted Mask [†] | Yes | 75.8 | 55.2 | 91.6 |
| + GT Mask (Ideal) | Yes | 82.5 | 69.2 | 92.7 |

[†] Overall predicted mask performance on `MD-3k`: mean IoU 0.8815.

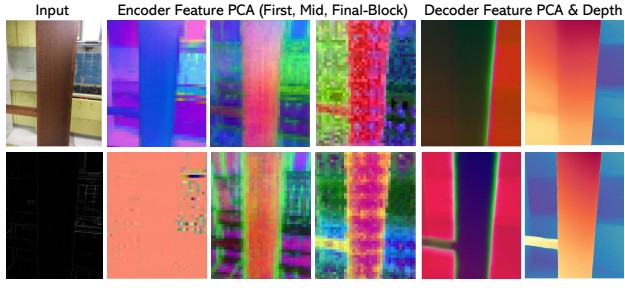

Figure 7: **Latent feature embedding comparison under RGB (Top) and LVP (Bottom) input on DAv2-L model.**

**Comparison to methods using extra semantic priors.** Table 2 shows that our training-free LVP approach with DAv2-L achieves an ML-SRA of 75.5%, which is highly competitive with the 75.8% from the semantic baseline. However, this baseline is not a simple alternative. As detailed in Appendix Sec. C.1 , it is a complex, multi-step pipeline that requires: 1) a separate, powerful, pre-trained segmentation model; 2) a specific depth model (DAv1-L) chosen for its known bias; and 3) a strong geometric assumption of planar surface interpolation. In contrast, our *single-model* LVP method achieves this comparable result with *no extra models, parameters, or such geometric assumptions* (e.g., our LVP-empowered method can handle curved surfaces, as shown in Fig. 12a). This highlights LVP's ability to elicit spatial structure as an emergent property directly from the frozen model's latent knowledge, without requiring additional semantic understanding.

**Visualization of LVP's internal mechanism.** To offer insight into why LVP succeeds, we visualize the DAv2-L model's latent features under both RGB and LVP inputs in Fig. 7. The visualization reveals that the two inputs create fundamentally different feature pathways starting from the earliest encoder blocks. While the standard RGB input (Top) generates features corresponding to the low-frequency foreground, the LVP prompt (Bottom) elicits distinct activations tied to the high-frequency background structure. This provides direct evidence that *LVP can steer model's latent representation*, compelling it to access and render a complementary hypothesis that was previously hidden.

### 4.3 ANALYSIS OF SCALING EFFECTS

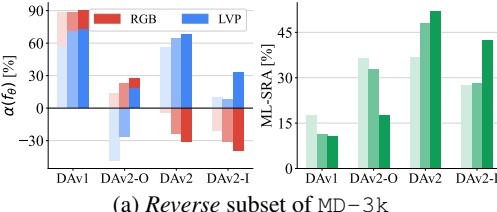
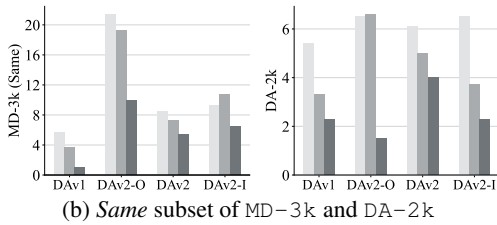

(a) *Reverse* subset of `MD-3k`    (b) *Same* subset of `MD-3k` and `DA-2k`

Figure 8: **Effect of model scale.** Small, Base, and Large variants shown with increasing shading color. (**a**) In ambiguous cases with reverse spatial relationship across layers, scaling improves ML-SRA [%] when RGB and LVP biases ($\alpha$) diverge. (**b**) In non-ambiguous cases, we present the performance gap between RGB and LVP, where larger models consistently generalize better.

**Model scaling effects are context-dependent in ambiguous multi-layer scenes.** For existing prevalent monocular depth foundation models, we find that simply increasing model size does not guarantee better multi-layer spatial understanding performance on ambiguous cases like transparent scenes. We observe two distinct patterns in Fig. 8a: **1**) For models where RGB and LVP inputs produce a similar bias (e.g., DAv1, DAv2-O both preferring the second layer), scaling up *hurts* ML-SRA. The model becomes more confident in a single hypothesis. **2**) For models where RGB and

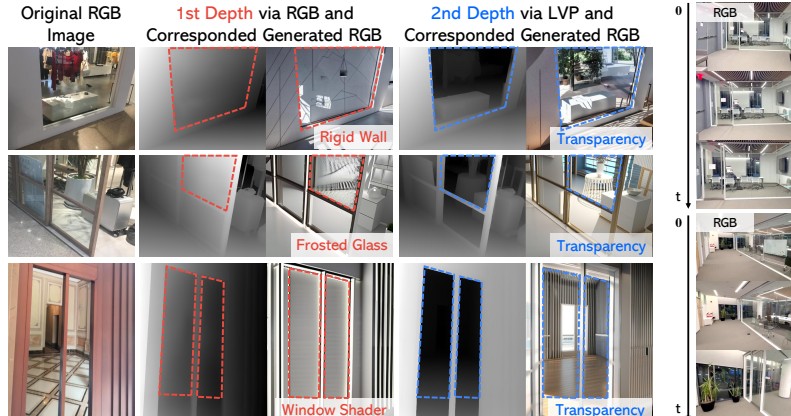

Figure 9: **Flexible geometry-conditioned image generation using decoupled two-layer depth, *i.e.*, transparent or opaque layer, separately.**

Figure 10: **Consistent two-layer depth estimation on monocular RGB video sequence.**

LVP produce opposite biases (e.g., DAv2, DAv2-I), scaling up *improves* ML-SRA. The larger model better separates the representations for the two depth layer hypotheses.

**Generalization in non-ambiguous multi-layer scenes.** Conversely, in non-ambiguous scenes, larger models consistently show better generalization, as evidenced by the shrinking performance gap between RGB and the out-of-distribution LVP input in Fig. 8b.

## 4.4 DOWNSTREAM APPLICATIONS

**Geometry-conditioned image generation.** As shown in Fig. 9, our multi-hypothesis depth maps provide flexible and comprehensive geometric control for generative models like ControlNet (Zhang et al., 2023b). This enables controllable and realistic synthesis of transparent or opaque environments.

**Temporally consistent video depth.** Fig. 10 shows that LVP enables temporally consistent bluetwo-layer depth for real-world video sequences with transparent objects. Both the foreground and background depth layers are stable across frames. For a clearer understanding, please refer to the **video demo** of two-layer depth prediction provided in the *supplementary material*.

## 4.5 ABLATION STUDY

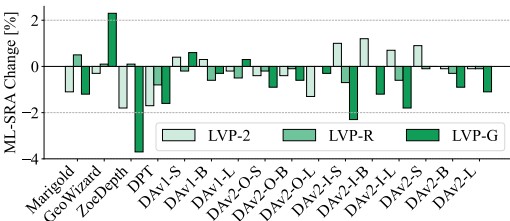

Figure 11: **Ablation of LVP design**. ML-SRA [%] change relative to the default LVP is shown.

Table 3: **LVP vs. Edge prompts.** ML-SRA [%] of DAv2 series on MD-3k *Reverse* subset. We evaluated over several low and high threshold pairs (e.g., Edge-1: (50, 150), Edge-2: (60, 180), Edge-3: (70, 210), and Edge-4: (80, 240)) that follow Canny Edge's recommendations.

| Method | Laplacian | Edge-1 | Edge-2 | Edge-3 | Edge-4 |
|--------|-----------|--------|--------|--------|--------|
| DAv2-S | **36.9** | 24.2 | 18.8 | 17.3 | 16.0 |
| DAv2-B | **48.2** | 39.8 | 32.4 | 30.6 | 28.1 |
| DAv2-L | **52.2** | 49.4 | 39.8 | 37.3 | 35.9 |

**Specific Laplacian visual prompt design.** As shown in Fig. 11, the performance remains robust across minor variations in the LVP design—such as different discretizations of the Laplacian kernel (4-neighbor vs. 8-neighbor, where LVP-2 uses the 8-neighbor, representing second-order differentiation) and sign adjustments (LVP-R uses a reversed sign compared to the default LVP). Even with a grayscale version (LVP-G), performance degrades only slightly. These results suggest that the core strength of LVP lies in its role as a ***high-pass spectral prompt***, rather than in the precise configuration of its kernel. This supports its use as a simple yet effective operator for spatial decoupling.

**Comparison to Canny edge prompts.** To analyze whether LVP is superior to just edge information, we compare LVP with a classic high-frequency prompt: the Canny edge map. As shown in Table 3, on the ambiguous *Reverse* subset, LVP outperforms all edge-only prompts for the best performing

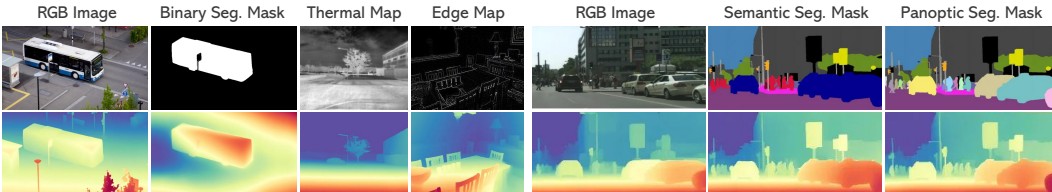

(a) Scenes with Curved Transparent Surfaces          (b) Scenes with Semi-Transparent Surfaces

Figure 12: **Two-layer depth prediction results from our LVP method (with DAv2-L baseline) on scenes with (a) curved and (b) semi-transparent surfaces.**

| RGB Image | Binary Seg. Mask | Thermal Map | Edge Map | RGB Image | Semantic Seg. Mask | Panoptic Seg. Mask |

Figure 13: **Generalization of depth prediction on non-ambiguous scenes to diverse high-frequency visual prompts beyond Laplacian input with base depth foundation model DAv2-L.**

model series, *i.e.*, DAv2, in Table 1. This suggests that the Laplacian's richer, non-binary derivative and intensity-related information provides a more robust prompt than a binary edge map.

**High-pass vs. low-pass visual prompts.** To test our mode-switching hypothesis, we perform an ablation study comparing our high-pass Laplacian prompt with a low-pass Gaussian prompt. As shown in Table 4, the Laplacian consistently elicits complementary depth layers and achieves strong ML-SRA scores on the *Reverse* subset, whereas the Gaussian prompt fails entirely. Conversely, for the non-ambiguous *Same* subset (results added as requested), both prompts achieve strong performance for some base depth foundation model settings. Notably, the Gaussian prompt achieves slightly higher scores on non-ambiguous cases, as it preserves the model's correct original depth hypothesis. This supports our hypothesis that a high-pass, not low-pass, perturbation is the key to switching modes in these ambiguous cases.

Table 4: **LVP vs Gaussian prompt on `MD−3k`.**

| Subset | (a) *Reverse* | | (b) *Same* | |
|---|---|---|---|---|
| Model | LVP | GAU | LVP | GAU |
| Marigold | **15.3** | 6.2 | 89.8 | **94.2** |
| GeoWizard | **17.6** | 6.0 | 91.9 | **95.3** |
| ZoeDepth | **45.4** | 0.7 | 86.8 | **94.6** |
| DPT | **46.4** | 0.3 | 88.7 | **94.3** |
| DAv1-S | **17.7** | 0.7 | 89.0 | **95.7** |
| DAv1-B | **11.4** | 0.3 | 91.5 | **96.1** |
| DAv1-L | **10.9** | 0.4 | 92.8 | **96.5** |
| DAv2-O-S | **36.6** | 5.7 | 83.5 | **92.6** |
| DAv2-O-B | **32.9** | 2.6 | 85.6 | **94.6** |
| DAv2-O-L | **17.6** | 0.6 | 93.4 | **96.4** |
| DAv2-I-S | **27.7** | 1.0 | 86.5 | **95.3** |
| DAv2-I-B | **28.1** | 0.7 | 91.2 | **96.6** |
| DAv2-I-L | **42.5** | 2.4 | 93.2 | **96.9** |
| DAv2-S | **36.9** | 3.6 | 90.7 | **97.8** |
| DAv2-B | **48.2** | 4.6 | 92.7 | **98.1** |
| DAv2-L | **52.2** | 4.5 | 93.6 | **98.2** |

## 5   CONCLUSION

We find that some depth foundation models harbor a surprisingly rich, multi-layered understanding of 3D scenes, often masked by training objectives that demand a single depth estimate. Using our `MD-3k` benchmark and Laplacian Visual Prompting (LVP) method, we can diagnose the depth layer preference bias for each model and unlock the latent two-layer spatial understanding for some models under ambiguous transparent scenes without retraining. Rather than pursuing ever-larger models or specialized training regimes, our work highlights a new perspective: ***eliciting more than one hypothesis from models otherwise forced into deterministic predictions.***

**Open questions.** While our study focuses on two-layer relative spatial understanding, extending this to metric depth estimation (Piccinelli et al., 2024; 2025b;a; Hu et al., 2024; Bochkovskii et al., 2024) and multi-depth blending are valuable next steps. Regarding limitations, LVP generalizes well to curved transparent surfaces (Fig. 12a) but struggles with semi-transparent or patterned glass (Fig. 12b), where disrupted high-frequency cues hinder isolation. Future work might leverage generative priors to resolve such occlusions. Most intriguingly, we observe that LVP elicits multiple hypotheses in ambiguous scenes while maintaining standard performance in non-ambiguous contexts (Fig. 13, Table Ac-d). This selective sensitivity suggests a broader robustness phenomenon. We invite the community to further explore these high-frequency prompting mechanisms of foundation models.

**Reproducibility statement.** To ensure the reproducibility of our findings and to encourage future research, ***our full source code used for implementation and analysis has been included in the supplementary material***. Upon publication, the benchmark and code will be made publicly available in a permanent repository.

**Ethics statement.** All authors have read and adhere to the ICLR Code of Ethics. This work ***does not involve human subjects or sensitive personal data***. All datasets are publicly available and properly cited, with no confidential information used. The methods and applications are intended for academic research, and potential risks of misuse have been considered. We declare no conflicts of interest and are committed to research integrity, fairness, and transparency.

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

| Baseline | (a) MD-3k (Overall) | | | | (b) MD-3k (Reverse) | | | | (c) MD-3k (Same) | | (d) DA-2k | |
|---|---|---|---|---|---|---|---|---|---|---|---|---|
| | RGB Input | | LVP Input | | RGB Input | | LVP Input | | RGB | LVP | RGB | LVP |
| | SRA(1) | SRA(2) | SRA(1) | SRA(2) | SRA(1) | SRA(2) | SRA(1) | SRA(2) | SRA(1/2) | SRA(1/2) | SRA | SRA |
| Random | 50.0 | 50.0 | 50.0 | 50.0 | 50.0 | 50.0 | 50.0 | 50.0 | 50.0 | 50.0 | 50.0 | 50.0 |
| Marigold | 70.7 | 79.9 | 70.8 | 77.4 | 39.6 | 60.4 | 42.3 | 57.7 | 95.0 | 92.6 | 88.9 | 78.9 |
| GeoWizard | 68.3 | 83.8 | 70.3 | 78.7 | 32.3 | 67.7 | 40.3 | 59.7 | 96.2 | 93.4 | 90.3 | 85.0 |
| ZoeDepth | 58.7 | 92.6 | 74.6 | 70.9 | 12.0 | 88.0 | 54.3 | 45.7 | 94.9 | 90.4 | 86.7 | 77.2 |
| DPT | 58.0 | 91.9 | 75.7 | 71.1 | 10.2 | 89.8 | 55.2 | 44.8 | 94.8 | 91.5 | 83.2 | 72.2 |
| Depth Pro | 84.9 | 69.6 | 73.5 | 76.8 | 67.6 | 32.4 | 46.2 | 53.8 | 98.3 | 94.6 | 95.8 | 84.1 |
| UniDepth-v2-L | 66.8 | 86.6 | 65.5 | 85.4 | 27.3 | 72.7 | 27.1 | 72.9 | 97.4 | 95.1 | 95.4 | 92.2 |
| UniK3D-L | 64.3 | 88.9 | 67.1 | 84.3 | 21.8 | 78.2 | 30.3 | 69.7 | 97.2 | 95.6 | 93.1 | 85.8 |
| DAv1-S | 56.8 | 95.3 | 60.6 | 85.4 | 5.8 | 94.2 | 21.5 | 78.5 | 96.1 | 90.7 | 88.7 | 83.0 |
| DAv1-B | 56.8 | 95.4 | 58.8 | 89.7 | 5.8 | 94.2 | 14.6 | 85.4 | 96.3 | 93.0 | 89.9 | 86.2 |
| DAv1-L | 56.6 | 96.0 | 59.2 | 90.9 | 4.8 | 95.2 | 13.6 | 86.4 | 96.6 | 94.3 | 89.5 | 88.5 |
| DAv2-O-S | 71.6 | 77.3 | 81.4 | 60.1 | 43.4 | 56.6 | 74.4 | 25.6 | 93.3 | 86.8 | 82.3 | 60.9 |
| DAv2-O-B | 70.3 | 80.3 | 77.2 | 65.8 | 38.5 | 61.5 | 63.1 | 36.9 | 94.8 | 88.2 | 89.7 | 70.4 |
| DAv2-O-L | 70.4 | 82.4 | 71.5 | 79.5 | 36.2 | 63.8 | 40.8 | 59.2 | 96.7 | 95.2 | 93.7 | 83.8 |
| DAv2-I-S | 80.4 | 71.2 | 70.0 | 74.3 | 60.6 | 39.4 | 45.1 | 54.9 | 95.8 | 89.3 | 88.5 | 79.2 |
| DAv2-I-B | 83.1 | 69.7 | 72.6 | 76.1 | 65.3 | 34.7 | 46.0 | 54.0 | 96.8 | 93.1 | 91.8 | 81.1 |
| DAv2-I-L | 85.2 | 68.1 | 68.1 | 82.6 | 69.6 | 30.4 | 33.5 | 66.5 | 97.3 | 95.0 | 94.8 | 88.4 |
| DAv2-S | 78.0 | 76.2 | 61.5 | 85.8 | 52.0 | 48.0 | 22.1 | 77.9 | 98.0 | 91.9 | 95.1 | 86.6 |
| DAv2-B | 82.4 | 72.3 | 60.5 | 88.6 | 61.7 | 38.3 | 17.7 | 82.3 | 98.5 | 93.5 | 96.7 | 89.5 |
| DAv2-L | 84.0 | 70.6 | 60.2 | 89.9 | 65.3 | 34.7 | 15.9 | 84.1 | 98.5 | 94.4 | 96.9 | 91.5 |

Table A: **Spatial Relationship Accuracy (SRA) [%] on (a-c) the MD-3k benchmark and (d) the DA-2k dataset (non-ambiguous reference).** SRA(1) and SRA(2) measure the accuracy between the model's single-layer depth estimate and the first and second layer annotation labels, respectively; higher values indicate a preference for that layer. Input modality (RGB or LVP) yielding higher SRA(1) or SRA(2) per model on the full *overall* set and *reverse* subset of MD-3k are **bolded**.

## A  MORE QUANTITATIVE RESULTS

**Detailed benchmarking results on MD-3k.** Table A presents per-layer Spatial Relationship Accuracy (SRA) [%] for various baseline models on the MD-3k benchmark and the DA-2k dataset (non-ambiguous reference) for future comparison.

**Detailed ablation study results of Laplacian Visual Prompting design on MD-3k.** Table B presents ML-SRA performance under different Laplacian discretization (4-neighbor vs. 8-neighbor in default LVP and LVP-2), kernel sign (LVP-R with reversed convolution vs. LVP), and color space (RGB vs. grayscale, LVP vs. LVP-G). The results show that ML-SRA performance remains largely unaffected, highlighting the crucial role of high-frequency information in 3D spatial decoupling.

## B  MORE QUALITATIVE RESULTS

**Multi-layer depth decoupling with Laplacian Visual Prompting.** In addition to demonstrating the effectiveness of LVP in revealing hidden depth layers in various models (Fig. A), we further demonstrate its capabilities using the best-performing baseline model, *i.e.*, the Depth Anything v2-Large (DAv2-L) model, in Figures C through H. LVP effectively elicits alternative depth hypotheses, particularly in scenes with transparency and occlusion. Although conventional depth estimations often fail to capture the layered nature of these scenes, collapsing multiple depths into a single layer, Laplacian-prompted depth maps reveal previously hidden depth layers, clearly delineating transparent surfaces and occluded objects behind the transparent surfaces.

**Failure cases in multi-layer depth decoupling with Laplacian Visual Prompting.** Despite significant improvements in multi-layer depth estimation, Laplacian Visual Prompting (LVP) is not immune to failure cases, especially due to its training-free nature. We present these failures in Fig. I. One type occurs when the initial single-layer depth prediction from the RGB input is already incorrect. In these instances, LVP struggles to correct the depth bias, resulting in inaccurate multi-layer depth

| Baseline | LVP (**Default**) | LVP-2 | LVP-R | LVP-G |
|---|---|---|---|---|
| Random | 25.0 | 25.0 | 25.0 | 25.0 |
| Marigold | 57.4 | 56.3 (-1.1) | 57.9 (+0.5) | 56.2 (-1.2) |
| GeoWizard | 59.5 | 59.2 (-0.3) | 59.6 (+0.1) | 61.8 (+2.3) |
| ZoeDepth | 68.8 | 67.0 (-1.8) | 68.9 (+0.1) | 65.1 (-3.7) |
| DPT | 70.2 | 68.5 (-1.7) | 69.4 (-0.8) | 68.6 (-1.6) |
| DAv1-S | 57.9 | 58.3 (+0.4) | 57.7 (-0.2) | 58.5 (+0.6) |
| DAv1-B | 56.6 | 56.9 (+0.3) | 56.0 (-0.6) | 56.3 (-0.3) |
| DAv1-L | 57.1 | 56.9 (-0.2) | 56.6 (-0.5) | 57.4 (+0.3) |
| DAv2-O-S | 63.0 | 62.6 (-0.4) | 62.8 (-0.2) | 62.1 (-0.9) |
| DAv2-O-B | 62.7 | 62.3 (-0.4) | 62.6 (-0.1) | 62.1 (-0.6) |
| DAv2-O-L | 60.4 | 59.1 (-1.3) | 60.4 (-0.0) | 60.1 (-0.3) |
| DAv2-I-S | 60.9 | 61.9 (+1.0) | 60.2 (-0.7) | 58.6 (-2.3) |
| DAv2-I-B | 63.7 | 64.9 (+1.2) | 63.7 (-0.0) | 62.5 (-1.2) |
| DAv2-I-L | 71.1 | 71.8 (+0.7) | 70.5 (-0.6) | 69.3 (-1.8) |
| DAv2-S | 67.2 | 68.1 (+0.9) | 67.1 (-0.1) | 67.2 (-0.0) |
| DAv2-B | 73.3 | 73.2 (-0.1) | 73.0 (-0.3) | 72.4 (-0.9) |
| DAv2-L | **75.5** | **75.4** (-0.1) | **75.4** (-0.1) | **74.4** (-1.1) |

Table B: **Ablation study of LVP design** via ML-SRA [%].

predictions. This is particularly problematic when the RGB model's depth map contains errors in ambiguous regions, limiting LVP's ability to produce accurate alternative depth hypotheses. The second type of failure happens when LVP predicts a depth layer similar to the RGB output, failing to decouple distinct depth layers. This occurs when LVP cannot extract sufficient high-frequency information from the Laplacian mask to differentiate between near and far surfaces, especially in scenes with low depth contrast or complex occlusions.

**Additional `MD-3k` benchmark samples.** Fig. J presents additional examples from the `MD-3k` benchmark, our newly introduced data set to evaluate the understanding of multi-layer spatial relationships. These examples highlight the diverse and challenging scenarios within `MD-3k`, including varying levels of depth ambiguity and transparency. By providing a broader range of scenes, we aim to assess how well models can disambiguate depth layers in multi-layered environments, particularly in real-world images that reflect the complexities and nuances of natural scenes.

## C  IMPLEMENTATION DETAILS

### C.1  MULTI-LAYER DEPTH ESTIMATION WITH SEMANTIC SEGMENTATION FOR COMPARISON

Our semantics-guided approach to multi-layer depth estimation integrates monocular depth predictions with semantic segmentation. We use the Depth Anything v1 Large model (DAv1-L) (Yang et al., 2024a) for initial single-layer depth estimation. As noted in the main paper, DAv1-L tends to predict greater depths in ambiguous regions. Building on this bias, we estimate the nearer depth layer, typically corresponding to transparent surfaces, by interpolating depth values from the boundaries of transparent regions. This process is guided by a segmentation mask of the transparent surface and informed by DAv1-L's depth estimates outside the ambiguous regions.

Figures K and L present qualitative results, showcasing both successful and failure cases. While this hybrid approach, combining DAv1-L's depth bias with semantic segmentation, achieves higher quantitative precision for multi-layer depth estimation than our training-free LVP method, we emphasize the importance of developing foundational models that can directly handle multi-layer depth estimation, rather than relying on task-specific model combinations.

## C.2 LAPLACIAN VISUAL PROMPTING FOR SEQUENTIAL VIDEO INFERENCE

We introduce Laplacian Visual Prompting (LVP) for sequential video inference, specifically for multi-layer depth estimation. Using the Video Depth Anything model as our baseline, LVP follows the multi-hypothesis depth estimation framework outlined in the main paper. Crucially, LVP enhances temporal consistency in depth estimation, mitigating inconsistencies common in per-frame processing. Additionally, the extracted hidden depth maps maintain high fidelity throughout the video sequence. A **video demonstration** is provided in the supplementary material.

## C.3 DEPTH-CONDITIONED IMAGE GENERATION

Our image generation pipeline uses depth maps and text prompts to guide scene synthesis. Depth maps are estimated using our proposed method, ensuring geometric accuracy, while text prompts modulate the visual attributes of the generated images.

We employ two distinct text prompts to control the scene's appearance:

- `a bright, well-lit photograph of an interior space with natural daylight, clear windows, balanced lighting, accurate geometry and structure, photorealistic, vibrant colors, modern interior design, clean and airy space`

The resulting RGB images, synthesized from multi-layer depth representations derived via LVP, are illustrated in Fig. M.

- `a bright, well-lit photograph of an interior space, accurate geometry and structure, photorealistic, modern interior design, clean and airy space`

The corresponding RGB images, generated using multi-layer depth representations from LVP, are presented in Fig. N.

These prompts enhance photorealistic scene synthesis by ensuring precise geometry and well-balanced lighting, complementing depth information. More details on the diffusion models used for depth-conditioned visual generation are available in the *Diffusers* library [1].

# D DATASHEET FOR `MD-3k` BENCHMARK

We document the necessary information about the proposed datasets and benchmarks following the guidelines of Gebru *et al.* (Gebru et al., 2021).

### D.1 MOTIVATION

Q1 **For what purpose was the dataset created?** Was there a specific task in mind? Was there a specific gap that needed to be filled? Please provide a description.

- Fig. B highlights the limitations of existing datasets for ambiguous transparent scenes. They often contain noisy raw depth from sensors (due to physical limitations) and inaccurate curated depth (due to human error). These challenges motivate our creation of the `MD-3k` benchmark.
- Our benchmark was created to evaluate multi-layer spatial perception, specifically focusing on the challenge of depth disentanglement in ambiguous 3D scenes. Existing depth datasets lack multi-layer spatial relationship labels, hindering fine-grained analysis in regions with transparency and spatial ambiguity. `MD-3k` fills this gap by providing the first benchmark with multi-layer spatial relationship labels, enabling detailed evaluation of models' ability to understand layered depth ordering.

Q2 **Who created the dataset (e.g., which team, research group) and on behalf of which entity (e.g., company, institution, organization)?**

---

[1]`https://huggingface.co/docs/diffusers/en/index`

- This benchmark is presented by [Anonymous Author][2]. Our aim is to advance the development and evaluation of spatial perception models, particularly in understanding multi-layer depth relationships in complex scenes.

Q3 **Who funded the creation of the dataset?** If there is an associated grant, please provide the name of the grantor and the grant name and number.

- This work was partially supported by [Anonymous Author].

Q4 **Any other comments?**

- No.

### D.2 COMPOSITION

Q5 **What do the instances that comprise the dataset represent (e.g., documents, photos, people, countries)?** *Are there multiple types of instances (e.g., movies, users, and ratings; people and interactions between them; nodes and edges)? Please provide a description.*

- The instances in `MD-3k` represent high-resolution RGB images of indoor and outdoor scenes containing ambiguous regions, particularly those involving transparent objects. Each instance is associated with segmentation masks highlighting ambiguous regions and pairwise spatial relationship labels for sparse points within these regions.

Q6 **How many instances are there in total (of each type, if appropriate)?**

- The `MD-3k` benchmark comprises 3,161 high-resolution RGB images. Each image contains annotations of spatial relationships for pairs of sparse points in ambiguous regions, totaling 3,161 annotated pairs. For each pair, two spatial relationship labels are provided, representing the relationship 'on' and another for the region 'behind' the transparent object.

Q7 **Does the dataset contain all possible instances or is it a sample (not necessarily random) of instances from a larger set?** *If the dataset is a sample, what is the larger set? Is the sample representative of the larger set (e.g., geographic coverage)? If so, please describe how this representativeness was validated/verified. If it is not representative of the larger set, please describe why not (e.g., to cover a more diverse range of instances, because instances were withheld or unavailable).*

- The dataset is a carefully selected sample from the GDD segmentation dataset (Mei et al., 2020). The larger set is the entire GDD dataset. The sample is not random but specifically chosen to include scenes rich in ambiguous regions, particularly those with transparent objects, to address the benchmark's focus on multi-layer spatial understanding in such challenging scenarios.

Q8 **What data does each instance consist of?** *"Raw" data (e.g., unprocessed text or images) or features? In either case, please provide a description.*

- Each instance consists of:
  - **RGB image:** High-resolution (720p) RGB image in PNG format.
  - **Segmentation masks:** Binary masks highlighting ambiguous regions within the RGB image, in PNG format.
  - **Spatial relationship labels:** Pairwise spatial relationship labels for sparse points in ambiguous regions, provided in JSON format. Each pair has two labels: one for the region 'on' and another for the region 'behind' the transparent object, indicating depth ordering (above and behind).

  This data is considered 'raw' in the sense that it is primarily image data and annotations, not pre-extracted features.

Q9 **Is there a label or target associated with each instance?** *If so, please provide a description.*

---

[2]Author and repository information have been anonymized in compliance with the double-blind policy and will be provided upon acceptance.

- Yes, the primary labels are the **pairwise spatial relationship labels**. For each annotated pair of sparse points in an ambiguous region of an RGB image, there are two labels indicating the spatial relationship (depth order) between the points in two layers: one for the region 'on' and another for the region 'behind' the transparent object. The labels are 'above' and 'behind'.

Q10 **Is any information missing from individual instances?** *If so, please provide a description, explaining why this information is missing (e.g., because it was unavailable). This does not include intentionally removed information, but might include, e.g., redacted text.*

- No, all instances are fully annotated with RGB images, segmentation masks, and spatial relationship labels.

Q11 **Are relationships between individual instances made explicit (e.g., users' movie ratings, social network links)?** *If so, please describe how these relationships are made explicit.*

- The instances are related by their source dataset, GDD (Mei et al., 2020). All images are selected from GDD and share the characteristics of scenes within that dataset. Furthermore, images are implicitly related by the common theme of containing ambiguous regions and transparent objects, as this was the selection criterion.

Q12 **Are there recommended data splits (e.g., training, development/validation, testing)?** *If so, please provide a description of these splits, explaining the rationale behind them.*

- No, we do not provide predefined data splits. Users are free to define their own splits based on their specific research needs. Common splits like training/validation/testing can be created randomly or based on scene characteristics if desired.

Q13 **Are there any errors, sources of noise, or redundancies in the dataset?** *If so, please provide a description.*

- We have implemented a rigorous annotation pipeline, including multi-round verification by expert annotators, to minimize errors and noise in the spatial relationship labels. However, as with any human annotation, there might be minor inconsistencies or subjective interpretations. We believe the overall quality of the annotations is high due to the careful curation process. Redundancies are not intentionally introduced.

Q14 **Is the dataset self-contained, or does it link to or otherwise rely on external resources (e.g., websites, tweets, other datasets)?** *If it links to or relies on external resources, a) are there guarantees that they will exist, and remain constant, over time; b) are there official archival versions of the complete dataset (i.e., including the external resources as they existed at the time the dataset was created); c) are there any restrictions (e.g., licenses, fees) associated with any of the external resources that might apply to a future user? Please provide descriptions of all external resources and any restrictions associated with them, as well as links or other access points, as appropriate.*

- The `MD-3k` benchmark is distributed as a self-contained dataset of annotations, segmentation masks, and image lists. It relies on the images from the GDD dataset (Mei et al., 2020) as the underlying visual data. Users will need to obtain the GDD dataset separately to use `MD-3k` fully.
  - a) We cannot guarantee the long-term availability of the GDD dataset. However, GDD is a publicly available dataset for research purposes.
  - b) We do not provide archival versions of the GDD dataset. Users should refer to the original GDD dataset sources for archival information.
  - c) Users should adhere to the licensing terms of the GDD dataset, which are separate from the `MD-3k` benchmark license. Please refer to the GDD dataset documentation for details on licenses and restrictions.

Q15 **Does the dataset contain data that might be considered confidential (e.g., data that is protected by legal privilege or by doctor–patient confidentiality, data that includes the content of individuals' non-public communications)?** *If so, please provide a description.*

- No, the `MD-3k` benchmark utilizes images from the publicly available GDD dataset, which does not contain confidential information.

Q16 **Does the dataset contain data that, if viewed directly, might be offensive, insulting, threatening, or might otherwise cause anxiety?** *If so, please describe why.*

- No, the images in the `MD-3k` benchmark depict common indoor and outdoor scenes and do not contain offensive, insulting, threatening, or anxiety-inducing content to the best of our knowledge.

Q17 **Does the dataset relate to people?** *If not, you may skip the remaining questions in this section.*

- No. This dataset primarily focuses on scenes and spatial relationships, and does not directly relate to people in a way that raises privacy or ethical concerns. While people may be present in some images as part of the general scene context, the annotations and benchmark tasks are not centered around individuals.

Q18 **Does the dataset identify any subpopulations (e.g., by age, gender)?**

- N/A.

Q19 **Is it possible to identify individuals (i.e., one or more natural persons), either directly or indirectly (i.e., in combination with other data) from the dataset?** *If so, please describe how.*

- N/A.

Q20 **Does the dataset contain data that might be considered sensitive in any way (e.g., data that reveals racial or ethnic origins, sexual orientations, religious beliefs, political opinions or union memberships, or locations; financial or health data; biometric or genetic data; forms of government identification, such as social security numbers; criminal history)?** *If so, please provide a description.*

- No.

Q21 **Any other comments?**

- We encourage users to exercise discretion and use the benchmark responsibly for research purposes only.

### D.3    COLLECTION PROCESS

Q22 **How was the data associated with each instance acquired?** *Was the data directly observable (e.g., raw text, movie ratings), reported by subjects (e.g., survey responses), or indirectly inferred/derived from other data (e.g., part-of-speech tags, model-based guesses for age or language)? If data was reported by subjects or indirectly inferred/derived from other data, was the data validated/verified? If so, please describe how.*

- The RGB images were directly observable, sourced from the GDD segmentation dataset (Mei et al., 2020). The segmentation masks and spatial relationship labels were indirectly derived through expert human annotation. Expert annotators manually identified ambiguous regions and provided pairwise spatial relationship labels. The annotations were validated through a multi-round verification process involving multiple annotators to ensure consistency and accuracy.

Q23 **What mechanisms or procedures were used to collect the data (e.g., hardware apparatus or sensor, manual human curation, software program, software API)?** *How were these mechanisms or procedures validated?*

- The data collection process primarily involved **manual human curation**. Expert annotators used in-house annotation tools to:
  - Visually inspect RGB images from the GDD dataset.
  - Identify and segment ambiguous regions, particularly those involving transparent objects, creating segmentation masks.
  - Select sparse point pairs within these ambiguous regions.
  - Determine and assign pairwise spatial relationship labels ('above' and 'behind') for each pair, considering both layers ('on' and 'behind' the transparent object).

The annotation procedure was validated through a multi-round verification process. Different annotators reviewed and cross-validated annotations to resolve discrepancies and ensure consistency and accuracy of the labels.

Q24 **If the dataset is a sample from a larger set, what was the sampling strategy (e.g., deterministic, probabilistic with specific sampling probabilities)?**

- The sampling strategy was **deterministic and targeted**. Images were selected from the GDD dataset based on a specific criterion: the presence of ambiguous regions, especially those featuring transparent objects. This was a deliberate selection process to focus the benchmark on the challenges of multi-layer spatial understanding in such complex scenes, rather than a random or probabilistic sampling of the entire GDD dataset.

Q25 **Who was involved in the data collection process (e.g., students, crowdworkers, contractors) and how were they compensated (e.g., how much were crowdworkers paid)?**

- The data collection process involved **expert annotators**. These were individuals with expertise in computer vision and image annotation, specifically trained for the task of identifying ambiguous regions and determining spatial relationships. Compensation details are proprietary to [Anonymous Author] and are not disclosed for anonymity purposes.

Q26 **Over what timeframe was the data collected? Does this timeframe match the creation timeframe of the data associated with the instances (e.g., recent crawl of old news articles)?** *If not, please describe the timeframe in which the data associated with the instances was created.*

- The data annotation and collection process took place between [Anonymous Month, Year] and [Anonymous Month, Year]. This timeframe represents the creation timeframe of the segmentation masks and spatial relationship labels associated with the images from the GDD dataset. The GDD dataset itself was created prior to this timeframe.

Q27 **Were any ethical review processes conducted (e.g., by an institutional review board)?** *If so, please provide a description of these review processes, including the outcomes, as well as a link or other access point to any supporting documentation.*

- Ethical review processes were not formally conducted by an institutional review board specifically for the creation of `MD-3k`. However, the benchmark utilizes publicly available images from the GDD dataset, which is intended for research purposes. We have taken care to ensure that the annotation process and the resulting benchmark do not raise ethical concerns regarding privacy or sensitive information.

Q28 **Does the dataset relate to people?** *If not, you may skip the remaining questions in this section.*

- No.

Q29 **Did you collect the data from the individuals in question directly, or obtain it via third parties or other sources (e.g., websites)?**

- N/A.

Q30 **Were the individuals in question notified about the data collection?** *If so, please describe (or show with screenshots or other information) how notice was provided, and provide a link or other access point to, or otherwise reproduce, the exact language of the notification itself.*

- N/A.

Q31 **Did the individuals in question consent to the collection and use of their data?** *If so, please describe (or show with screenshots or other information) how consent was requested and provided, and provide a link or other access point to, or otherwise reproduce, the exact language to which the individuals consented.*

- N/A.

Q32 **If consent was obtained, were the consenting individuals provided with a mechanism to revoke their consent in the future or for certain uses?** *If so, please provide a description, as well as a link or other access point to the mechanism (if appropriate).*

- N/A.

**Q33 Has an analysis of the potential impact of the dataset and its use on data subjects (e.g., a data protection impact analysis) been conducted?** *If so, please provide a description of this analysis, including the outcomes, as well as a link or other access point to any supporting documentation.*

- No formal data protection impact analysis has been conducted. However, as the dataset does not directly relate to people and uses publicly available images, we anticipate minimal risk to data subjects. The focus of the benchmark is on evaluating computer vision models for spatial understanding. We encourage responsible use of the benchmark for research purposes.

**Q34 Any other comments?**

- No.

### D.4    PREPROCESSING, CLEANING, AND/OR LABELING

**Q35 Was any preprocessing/cleaning/labeling of the data done (e.g., discretization or bucketing, tokenization, part-of-speech tagging, SIFT feature extraction, removal of instances, processing of missing values)?** *If so, please provide a description. If not, you may skip the remainder of the questions in this section.*

- Yes, **labeling** was performed. Expert annotators manually labeled spatial relationships for pairs of sparse points in ambiguous regions. This labeling process is the core contribution of the `MD-3k` benchmark. No other preprocessing or cleaning of the RGB images from the GDD dataset was performed.

**Q36 Was the "raw" data saved in addition to the preprocessed/cleaned/labeled data (e.g., to support unanticipated future uses)?** *If so, please provide a link or other access point to the "raw" data.*

- N/A. The 'raw' data in this context would be the original RGB images from the GDD dataset. We are distributing the segmentation masks and spatial relationship labels, which are the 'labeled' data. The 'raw' RGB images are available from the original GDD dataset (Mei et al., 2020).

**Q37 Is the software used to preprocess/clean/label the instances available?** *If so, please provide a link or other access point.*

- The in-house annotation tools used for labeling are not publicly released at this time due to proprietary reasons. However, we will provide detailed descriptions of the annotation process and data format to facilitate the use of the benchmark.

**Q38 Any other comments?**

- No.

### D.5    USES

**Q39 Has the dataset been used for any tasks already?** *If so, please provide a description.*

- Not yet. `MD-3k` is a newly introduced benchmark. We will present initial baseline evaluations in our paper.

**Q40 Is there a repository that links to any or all papers or systems that use the dataset?** *If so, please provide a link or other access point.*

- We will maintain a repository at [Anonymous Repo] that links to papers and systems that utilize the `MD-3k` benchmark as they become available.

**Q41 What (other) tasks could the dataset be used for?**

- The primary intended use of `MD-3k` is for evaluating models for **multi-layer spatial understanding** and **depth disentanglement** in ambiguous scenes. Specifically, it can be used to:
  - Evaluate the performance of depth estimation models in regions with transparency and complex spatial arrangements.

- Benchmark algorithms designed for understanding layered scene representations.
- Analyze the ability of models to reason about relative depth ordering in multi-layer contexts.
- Develop and test novel approaches for handling spatial ambiguity in 3D scene understanding.

It can also be used for related tasks such as transparent object segmentation and reasoning about spatial relationships in general.

**Q42 Is there anything about the composition of the dataset or the way it was collected and preprocessed/cleaned/labeled that might impact future uses?** *For example, is there anything that a future user might need to know to avoid uses that could result in unfair treatment of individuals or groups (e.g., stereotyping, quality of service issues) or other undesirable harms (e.g., financial harms, legal risks)? If so, please provide a description. Is there anything a future user could do to mitigate these undesirable harms?*

- The `MD-3k` benchmark is focused on ambiguous scenes, particularly those with transparent objects. Users should be aware that the dataset is specifically designed to challenge models in these scenarios. It might not be representative of general scenes without ambiguity. Future users should consider this focus when applying the benchmark and interpreting results. As the dataset does not relate to people or sensitive attributes, the risk of unfair treatment or other harms is considered low. However, responsible and ethical use of the benchmark is always encouraged.

**Q43 Are there tasks for which the dataset should not be used?** *If so, please provide a description.*

- We are not aware of any specific tasks for which `MD-3k` should not be used. However, its primary focus is on multi-layer spatial understanding in ambiguous regions. Using it for tasks completely unrelated to spatial reasoning or depth perception might not be appropriate.

**Q44 Any other comments?**

- No.

### D.6   DISTRIBUTION AND LICENSE

**Q45 Will the dataset be distributed to third parties outside of the entity (e.g., company, institution, organization) on behalf of which the dataset was created?** *If so, please provide a description.*

- Yes, the `MD-3k` benchmark will be publicly distributed to third parties for research purposes.

**Q46 How will the dataset be distributed (e.g., tarball on website, API, GitHub)?** *Does the dataset have a digital object identifier (DOI)?*

- The `MD-3k` benchmark, including annotations, segmentation masks, and code for evaluation, will be distributed as a downloadable tarball via [Anonymous Repo], likely a GitHub repository. We plan to obtain a DOI for the benchmark upon publication of the associated paper.

**Q47 When will the dataset be distributed?**

- We plan to distribute the dataset publicly starting from March, 2025, and onwards, coinciding with the anticipated publication of our work.

**Q48 Will the dataset be distributed under a copyright or other intellectual property (IP) license, and/or under applicable terms of use (ToU)?** *If so, please describe this license and/or ToU, and provide a link or other access point to, or otherwise reproduce, any relevant licensing terms or ToU, as well as any fees associated with these restrictions.*

- The `MD-3k` benchmark, including annotations and code, will be released under the **Apache-2.0 license**. This is an open-source license that allows for free use, modification, and distribution for research and commercial purposes, with proper attribution. The full license text will be available in the benchmark repository at [Anonymous Repo]. There are no fees associated with the use of the `MD-3k` benchmark under this license.

Q49 **Have any third parties imposed IP-based or other restrictions on the data associated with the instances?** *If so, please describe these restrictions, and provide a link or other access point to, or otherwise reproduce, any relevant licensing terms, as well as any fees associated with these restrictions.*

- The `MD-3k` benchmark relies on RGB images from the GDD dataset (Mei et al., 2020). Users of `MD-3k` should also comply with the licensing terms of the GDD dataset, which are separate from the Apache-2.0 license of our benchmark. We recommend users refer to the GDD dataset documentation for details on their specific licensing terms and any potential restrictions. We are not aware of any IP-based or other restrictions imposed by third parties directly on our annotations and benchmark data, other than the underlying GDD images.

Q50 **Do any export controls or other regulatory restrictions apply to the dataset or to individual instances?** *If so, please describe these restrictions, and provide a link or other access point to, or otherwise reproduce, any supporting documentation.*

- No, we are not aware of any export controls or other regulatory restrictions applicable to the `MD-3k` benchmark or its individual instances.

Q51 **Any other comments?**

- No.

### D.7 MAINTENANCE

Q52 **Who will be supporting/hosting/maintaining the dataset?**

- Anonymous Author will be responsible for supporting, hosting, and maintaining the `MD-3k` benchmark.

Q53 **How can the owner/curator/manager of the dataset be contacted (e.g., email address)?**

- The owner/curator/manager of the dataset can be contacted through [Anonymous Contact Method], which will be provided in the benchmark repository at [Anonymous Repo] upon release.

Q54 **Is there an erratum?** *If so, please provide a link or other access point.*

- There is no erratum for the initial release of `MD-3k`. Any errata identified in the future will be documented and communicated through the benchmark repository at [Anonymous Repo].

Q55 **Will the dataset be updated (e.g., to correct labeling errors, add new instances, delete instances)?** *If so, please describe how often, by whom, and how updates will be communicated to users (e.g., mailing list, GitHub)?*

- Yes, we plan to update the `MD-3k` benchmark periodically. Updates may include corrections of labeling errors, addition of new instances, or improvements to the benchmark resources. Updates will be performed by [Anonymous Author] and will be communicated to users through the benchmark repository at [Anonymous Repo], potentially via release notes and/or announcements on the repository's issue tracker.

Q56 **If the dataset relates to people, are there applicable limits on the retention of the data associated with the instances (e.g., were individuals in question told that their data would be retained for a fixed period of time and then deleted)?** *If so, please describe these limits and explain how they will be enforced.*

- N/A.

Q57 **Will older versions of the dataset continue to be supported/hosted/maintained?** *If so, please describe how. If not, please describe how its obsolescence will be communicated to users.*

- We intend to host and maintain older versions of the `MD-3k` benchmark in the benchmark repository at [Anonymous Repo], likely through version control mechanisms (e.g., Git tags or branches). This will allow users to access and utilize specific versions of the benchmark for reproducibility and comparison purposes. If a version becomes obsolete, it will be clearly marked as such in the repository, but will remain accessible.

Q58 **If others want to extend/augment/build on/contribute to the dataset, is there a mechanism for them to do so?** *If so, please provide a description. Will these contributions be validated/verified? If so, please describe how. If not, why not? Is there a process for communicating/distributing these contributions to other users? If so, please provide a description.*

- We welcome contributions from the community to extend, augment, or build upon the `MD-3k` benchmark. Users can contribute by:
  - Reporting issues or suggesting improvements via the issue tracker in the benchmark repository at [Anonymous Repo].
  - Submitting pull requests with code contributions (e.g., evaluation scripts, new baselines).
  - Proposing new annotations or extensions to the dataset by contacting [Anonymous Author] through the contact method provided in the repository.

Q59 **Any other comments?**

- No.

## E  MORE RELATED WORK

### E.1  VISUAL PROMPTING AND MODEL INTERROGATION

Adapting large, pre-trained vision models without costly full fine-tuning has become a critical area of study (Ding et al., 2024; Zhang et al., 2025). The dominant paradigm is Parameter-Efficient Fine-Tuning (PEFT), where a small set of *learnable* parameters are trained. This includes methods like adapters, which insert lightweight modules into the model (Houlsby et al., 2019), and Visual Prompt Tuning (VPT), which introduces learnable tokens into the input space to steer a frozen backbone toward a new downstream task (Jia et al., 2022). In contrast, our work belongs to the **training-free, input-space prompting** paradigm, which modifies the raw input pixels at test time, requiring no gradient-based optimization. This category itself has two distinct goals. The first is *task adaptation or control*. Bahng et al. first explored this by using fixed, non-learned patches or inpainting to adapt models for various tasks (Bahng et al., 2022). This concept has been powerfully extended in generative models like ControlNet, which uses spatial conditioning maps (e.g., edges, depth) as a strong prompt to guide image generation (Zhang et al., 2023b), and in recent work like Motion Prompting, which uses user-drawn trajectories to control video synthesis (Geng et al., 2025). The second, more nascent goal is *model interrogation*: probing a model to understand its internal representations. A recent work investigates multi-modal LLMs to generate cognitive maps to visualize their latent spatial reasoning (Yang et al., 2025).

**Our Focus:** Our Laplacian Visual Prompting (LVP) is a novel form of training-free interrogation. Like Motion Prompting (Geng et al., 2025), we use a structured, non-learned input; however, like Thinking in Space (Yang et al., 2025), our goal is to *interrogate* latent 3D knowledge. Our novelty lies in using a classical differential operator (the Laplacian) as a visual question to *decouple* a model's ambiguous geometric hypotheses. We are not adapting the model to a new task but rather revealing a deeper, multi-layer capability it already possesses.

### E.2  MULTI-HYPOTHESIS SPATIAL UNDERSTANDING

Monocular 3D perception is a fundamentally ill-posed problem, as a single 2D image can correspond to many valid 3D scenes. This ambiguity is a long-standing challenge (Hariat et al., 2025). Historically, this ambiguity was resolved using additional information. **Multi-View Stereo (MVS)** (Campbell et al., 2008; Yao et al., 2018) samples discrete depth hypotheses (i.e., depth planes) and fuses information across multiple views to find the single most photo-consistent 3D structure (Peng et al., 2022). In other domains like 3D human pose, the ambiguity is resolved by designing *specialized architectures*. Because a single 2D pose can correspond to many valid 3D poses, methods are explicitly designed to predict a *distribution* or a *discrete set* of plausible 3D poses using generative models or neural distance fields (Chen et al., 2023c; Bae et al., 2023). This challenge is particularly acute in **monocular depth estimation**. Recent foundation models have achieved remarkable zero-shot performance. However, they are still trained to produce a single, deterministic depth map. To handle

ambiguity, recent work has explored several new directions. First, specialized multi-layer models are being designed; this requires *training specialized, supervised models* for multi-layer output (Hariat et al., 2025). Second, generative and probabilistic models are being explored. Diffusion models and other probabilistic frameworks can capture a distribution of depth values, implicitly modeling ambiguity (Bae et al., 2023). Third, some work uses input augmentation *during training*. Hariat et al. (2025) use a Distance Transform—another classical operator—to augment inputs during self-supervised training. Their goal, however, is to *resolve* the ambiguity and produce a single, more robust depth map (Hariat et al., 2025).

**Our Focus:** Our approach is fundamentally different from all prior work. We do not (**a**) require multiple views, (**b**) design a new specialized architecture, (**c**) use a generative model, or (**d**) augment data during a new training phase. We are the first to demonstrate that **standard, frozen, single-depth foundation models** (e.g., DPT, Depth Anything) *already* contain the latent knowledge to perceive multi-layer scenes. Our contribution is an inference-time, training-free method to *decouple and elicit* these distinct hypotheses, rather than training a new model or collapsing the ambiguity into a single, resolved output.

### E.3 EMERGING PROPERTIES FOR VISUAL MODELS

A growing body of work shows that large visual foundation models trained with generative or geometric pretext objectives can naturally encode dense pixel correspondences without explicit supervision. On the generative side, intermediate activations from Stable Diffusion's U-Net architecture can directly serve as training-free descriptors for semantic matching, achieving impressive zero-shot performance on standard correspondence benchmarks (Tang et al., 2023b). Subsequent studies have improved robustness by aggregating features across diffusion timesteps and layers (Luo et al., 2023), or by combining diffusion features with complementary self-supervised representations (Zhang et al., 2023a). In parallel, approaches based on cross-view completion have explicitly fostered correspondence emergence. CroCo v2, trained to reconstruct a target view from a source view, produces internal cross-attention maps that effectively function as zero-shot correspondence fields, removing the need for task-specific training (Weinzaepfel et al., 2023; An et al., 2025). Collectively, these findings suggest a convergence where both generative and geometric pretraining strategies can yield training-free correspondences through distinct yet complementary mechanisms.

Moving beyond purely 2D matching, recent work explores latent 3D and 4D structures encoded by models pretrained solely on 2D data. Studies show that self-supervised and generative backbones can linearly encode single-view depth and surface normals, whereas vision-language encoders typically demonstrate weaker representations of these geometric attributes (El Banani et al., 2024). Building upon powerful static 3D point-map backbones, such as DUSt3R (Wang et al., 2024b) and its dynamic extension MonST3R (Zhang et al., 2024b), a prominent research direction develops training-free adapters usable during inference. Easi3R exemplifies this trend by aggregating DUSt3R's cross-attention patterns spatially and temporally to generate motion masks per frame. By re-weighting attention in a subsequent forward pass, Easi3R cleanly separates camera motion from object motion, recovering temporally coherent dynamic point maps and poses without additional fine-tuning (Chen et al., 2025a). Concurrently, generative approaches leveraging diffusion priors enable training-free or test-time-guided multi-view-consistent rendering and 3D representation, as demonstrated by methods such as Zero-1-to-3 (Liu et al., 2023), ViewFusion (Yang et al., 2024c), ZeroNVS (Sargent et al., 2024), and DreamGaussian (Tang et al., 2023a). Meanwhile, stronger foundation models and specialized probes provide clearer insights into the capabilities of training-free adapters. Notable examples include MASt3R, which uses 3D structure to regularize correspondence (Leroy et al., 2024); CUT3R, which demonstrates persistent-state forward 3D and 4D perception (Wang et al., 2025); and Feat2GS, which probes geometric and textural understanding of visual foundation models via Gaussian splatting (Chen et al., 2025b). Additionally, emergent 3D semantic capabilities are evidenced by approaches such as LERF, which integrates language priors into radiance fields (Kerr et al., 2023), and SA3D, which generalizes 2D segmentation priors into 3D segmentation tasks (Cen et al., 2025). Together, these results indicate that large-scale 2D pretraining implicitly embeds rich semantic-geometric structures accessible without further task-specific training.

A complementary direction investigates modern video generation models as prompt-driven visual reasoning systems capable of solving diverse tasks without additional training. Wiedemer et al. systematically evaluated Veo-3, revealing its strong zero-shot performance across a wide range of

tasks, from basic perception and physically grounded modeling to manipulation tasks and early visual reasoning problems such as maze-solving and visual analogies (Wiedemer et al., 2025). They propose a chain-of-frames reasoning approach analogous to the chain-of-thought strategy commonly used in language models. This direction aligns well with recent progress in video diffusion modeling (Blattmann et al., 2023) and findings demonstrating that diffusion models can function as zero-shot classifiers when suitably probed (Clark & Jaini, 2023). Moreover, temporally consistent depth estimation has been extracted from video diffusion priors with minimal additional training, highlighting their potential as sources of 4D geometric information at test time (Shao et al., 2025). Within this broader landscape, training-free 3D systems such as LVP can be positioned along two primary axes. First, geometry can either be extracted directly from internal model representations, such as through attention maps as in Easi3R (Chen et al., 2025a), or guided explicitly via constraints injected into generative processes. Second, methods must balance precise, metric, physically accurate 3D reconstruction against versatile, general-purpose models that often sacrifice some geometric fidelity in exchange for broader applicability.

## F  LIMITATION AND FUTURE WORK

As a training-free probe, LVP's effectiveness depends heavily on the representational capacity of the underlying base model and the presence of sufficient high-frequency content in the scene. It may fail when the initial RGB-based depth prediction is fundamentally flawed or when the scene lacks sufficient texture for the operator to amplify. Additionally, LVP yields a deterministic alternative hypothesis rather than a probabilistic one, limiting its ability to represent the full posterior distribution.

Future work will address these limitations. A primary direction is to integrate LVP with uncertainty estimation frameworks to generate probabilistic, multi-hypothesis depth maps suitable for safety-critical applications. We also aim to generalize the approach by constructing a dictionary of input-space interrogations, incorporating both non-learned (e.g., wavelet-based, Hessian-based) and learned adaptive prompts (Wu et al., 2024) to probe other latent phenomena, such as reflections. Ultimately, our findings suggest that interrogating representational diversity may offer a more robust path toward spatial understanding than merely scaling model size.

## G  BROADER IMPACT

This work transcends monocular depth estimation, tackling the fundamental challenge of depth ambiguity that limits spatial understanding across AI. By resolving this core limitation, we pave the way for more reliable and versatile AI in complex 3D environments, impacting diverse fields reliant on robust perception.

Our central contribution, Laplacian Visual Prompting (LVP), introduces a training-free spectral technique. LVP empowers models to explicitly disentangle depth ambiguity and generate multi-hypothesis predictions, unlocking latent spatial knowledge. This broadly applicable spectral prompting paradigm extends beyond depth, offering a transformative tool for visual model adaptation and interpretation across tasks grappling with ambiguity and layered representations.

The `MD-3k` benchmark, the first of its kind with multi-layer spatial relationship labels, provides a critical platform for rigorous evaluation of multi-layer spatial understanding. `MD-3k` enables fine-grained analysis of depth disentanglement, pushing beyond single-layer metrics and driving progress in foundational spatial intelligence for computer vision, robotics, and AI.

Ultimately, LVP demonstrates spectral prompting as a powerful mechanism to unlock zero-shot multi-layer spatial understanding from existing models. This paradigm shift towards spectral prompting and multi-hypothesis spatial foundation models opens transformative avenues for interpretable, adaptable, and robust AI. By conquering depth ambiguity, this research delivers essential tools and insights for building foundational models capable of truly understanding real-world 3D scenes.

## H  AVAILABILITY AND MAINTENANCE

To accelerate progress in multi-layer spatial understanding and spectral prompting, all code and datasets from this study are publicly released at [Anonymous Repo]. This repository provides:

- **Laplacian Visual Prompting (LVP) code**. Ready-to-use implementation for spectral prompting.
- **MD-3k benchmark**. The first multi-layer spatial relationship dataset for rigorous evaluation.
- **Evaluation suite and baselines**. Code to reproduce and extend our experimental results.
- **Reproduction guide**. Step-by-step instructions for full experiment replication.

We are committed to the sustained accessibility and usability of these resources, empowering the community to build upon this foundation and drive future innovations in ambiguity-free spatial AI.

## I    LICENSE

The MD-3k benchmark and the Laplacian Visual Prompting code are released under the Apache License 2.0.

## J    LLM USAGE STATEMENT

We used LLM to proofread the text and improve sentence clarity.

## K    PUBLIC RESOURCES USED

We acknowledge the following public resources used in this work:

- Depth-Anything-v2[3] .................................... Apache-2.0+CC-BY-NC-4.0
- Depth-Anything[4] ................................................... Apache-2.0
- DPT[5] ..................................................................... MIT
- ZoeDepth[6] ............................................................... MIT
- Marigold[7] ........................................................ Apache-2.0
- GeoWizard[8] ......................................................... CC BY 4.0
- Diffusers[9] ....................................................... Apache-2.0
- Video Depth Anything[10] .......................................... Apache-2.0

---

[3] https://github.com/DepthAnything/Depth-Anything-V2
[4] https://github.com/LiheYoung/Depth-Anything.
[5] https://github.com/isl-org/DPT.
[6] https://github.com/isl-org/ZoeDepth.
[7] https://github.com/prs-eth/Marigold.
[8] https://github.com/fuxiao0719/GeoWizard.
[9] https://github.com/huggingface/diffusers/tree/main
[10] https://github.com/DepthAnything/Video-Depth-Anything

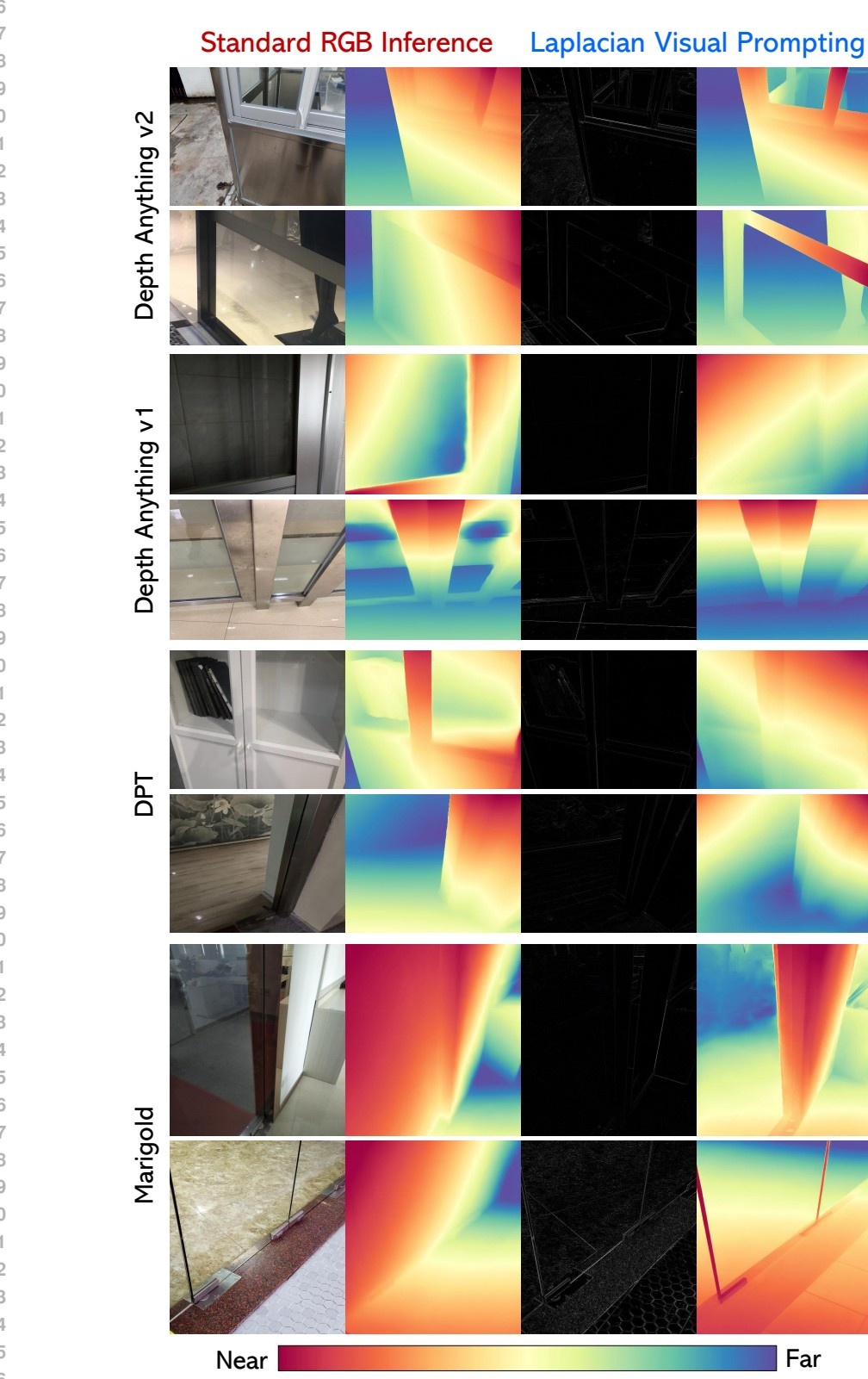

Figure A: **Unlocking *hidden depth* with Laplacian Visual Prompting on diverse models (**Yang et al., 2024a;b; Ke et al., 2024; Ranftl et al., 2021**).** Each case shows the RGB input, the estimated depth via RGB, the Laplacian input, and the estimated *depth* via Laplacian input.

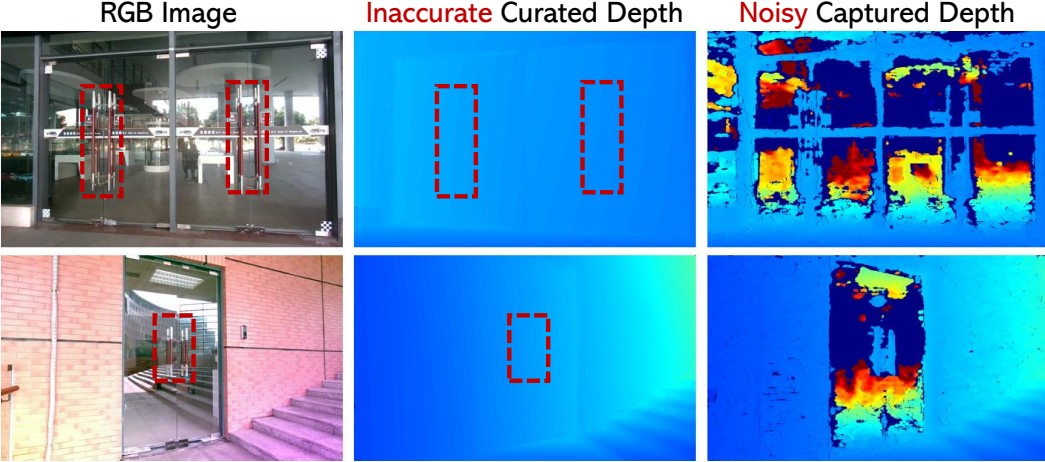

Figure B: Noise and inaccuracies in depth data from existing datasets (Liang et al., 2023) for complex, ambiguous scenes. These arise from limitations in both sensor acquisition and human annotation.

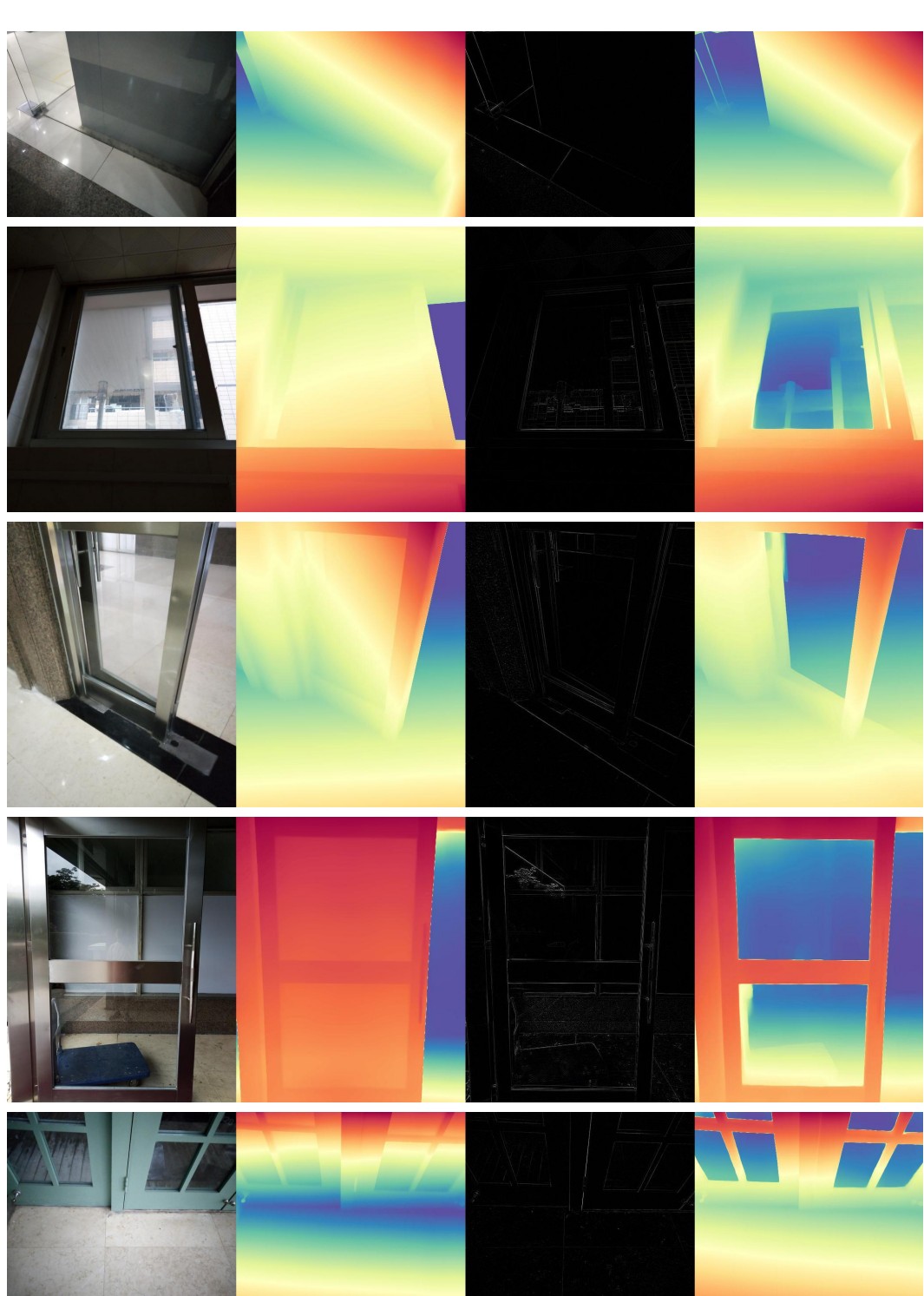

Figure C: **LVP-empowered multi-layer depth.** Each case includes an RGB image with its depth, and the Laplacian with its depth.

1728
1729
1730
1731
1732
1733
1734
1735
1736
1737
1738
1739
1740
1741
1742
1743
1744
1745
1746
1747
1748
1749
1750
1751
1752
1753
1754
1755
1756
1757
1758
1759
1760
1761
1762
1763
1764
1765
1766
1767
1768
1769
1770
1771
1772
1773
1774
1775
1776
1777
1778
1779
1780
1781

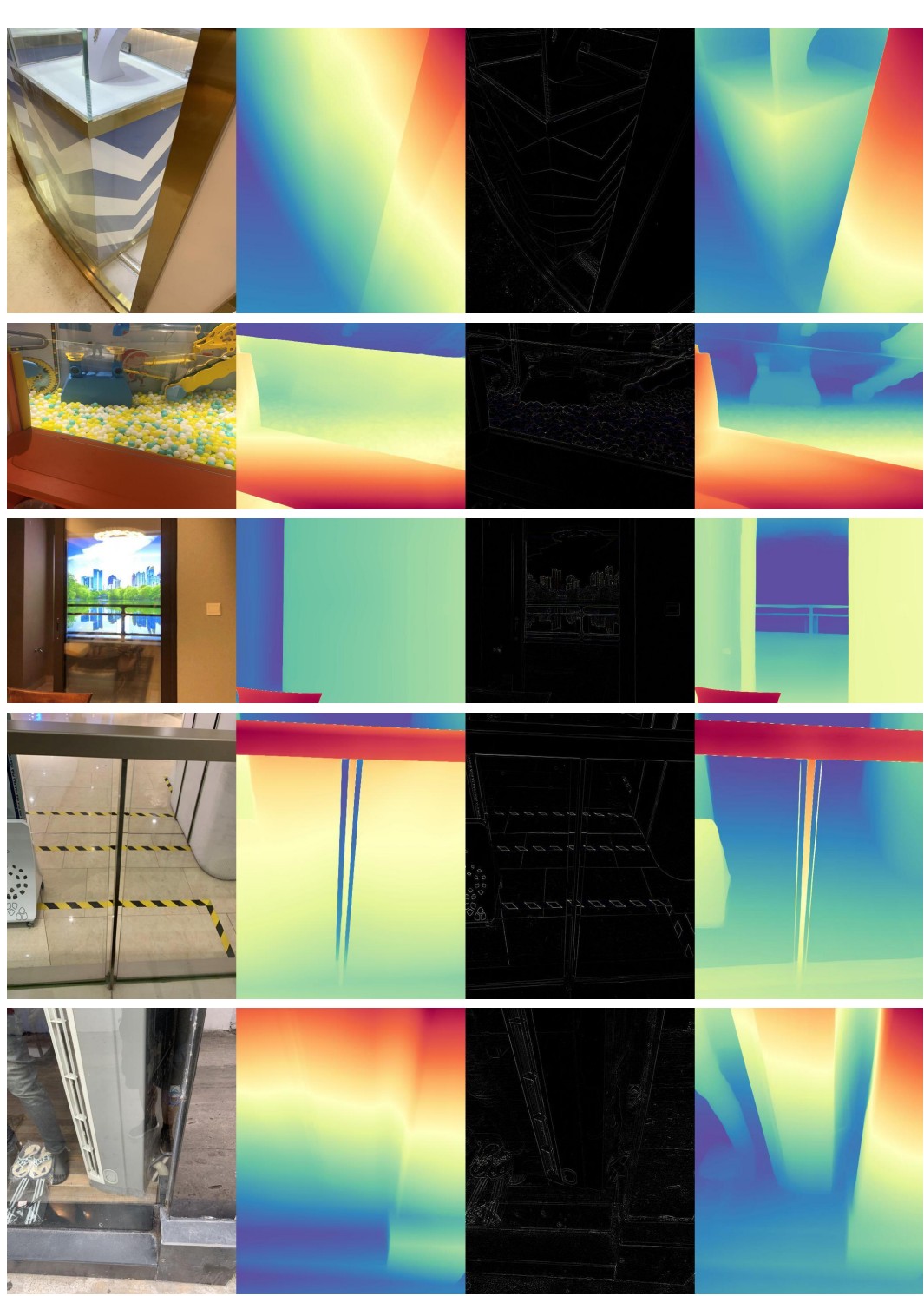

Figure D: **LVP-empowered multi-layer depth.** Each case includes an RGB image with its depth, and the Laplacian with its depth.

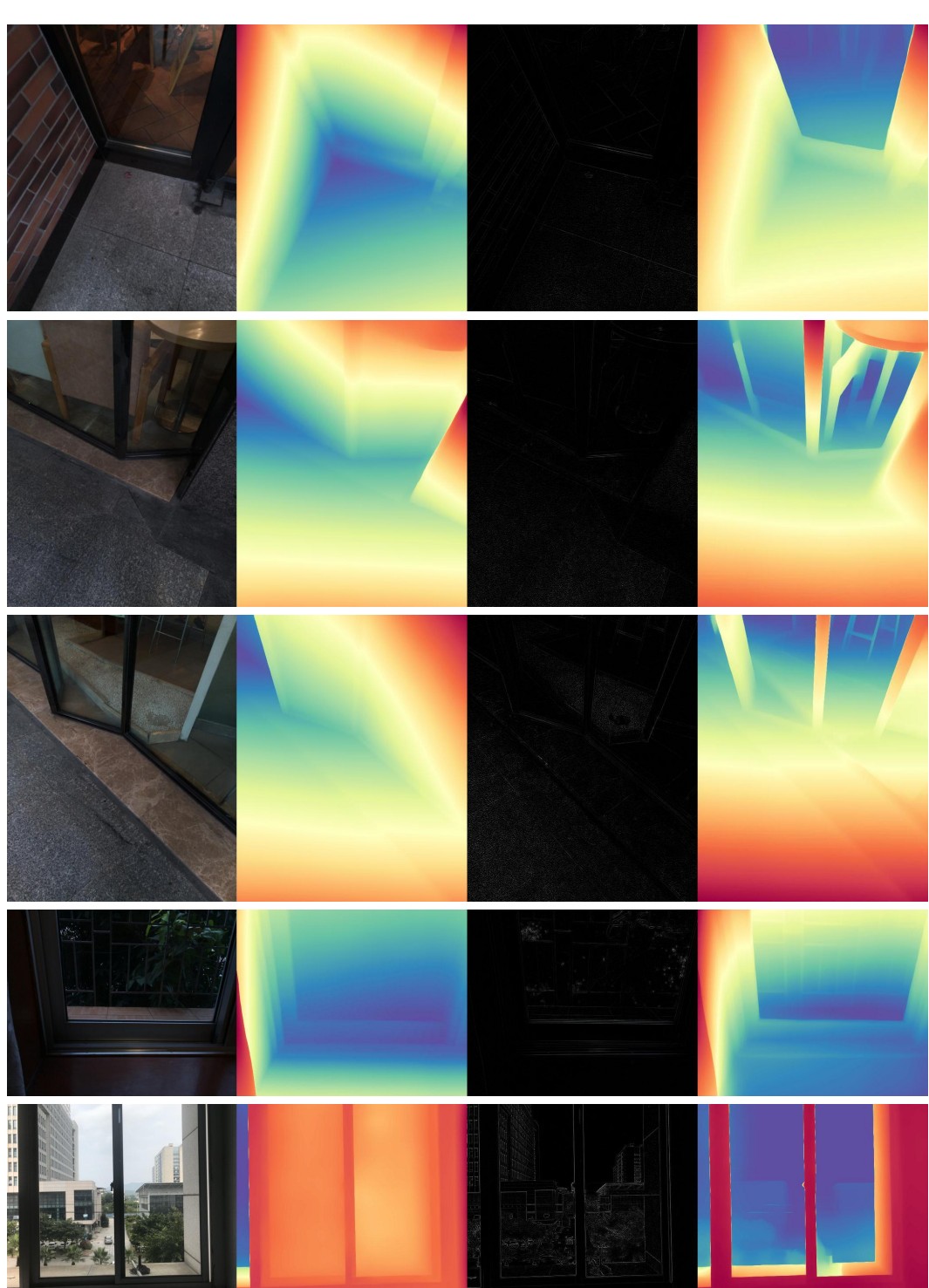

Figure E: **LVP-empowered multi-layer depth.** Each case includes an RGB image with its depth, and the Laplacian with its depth.

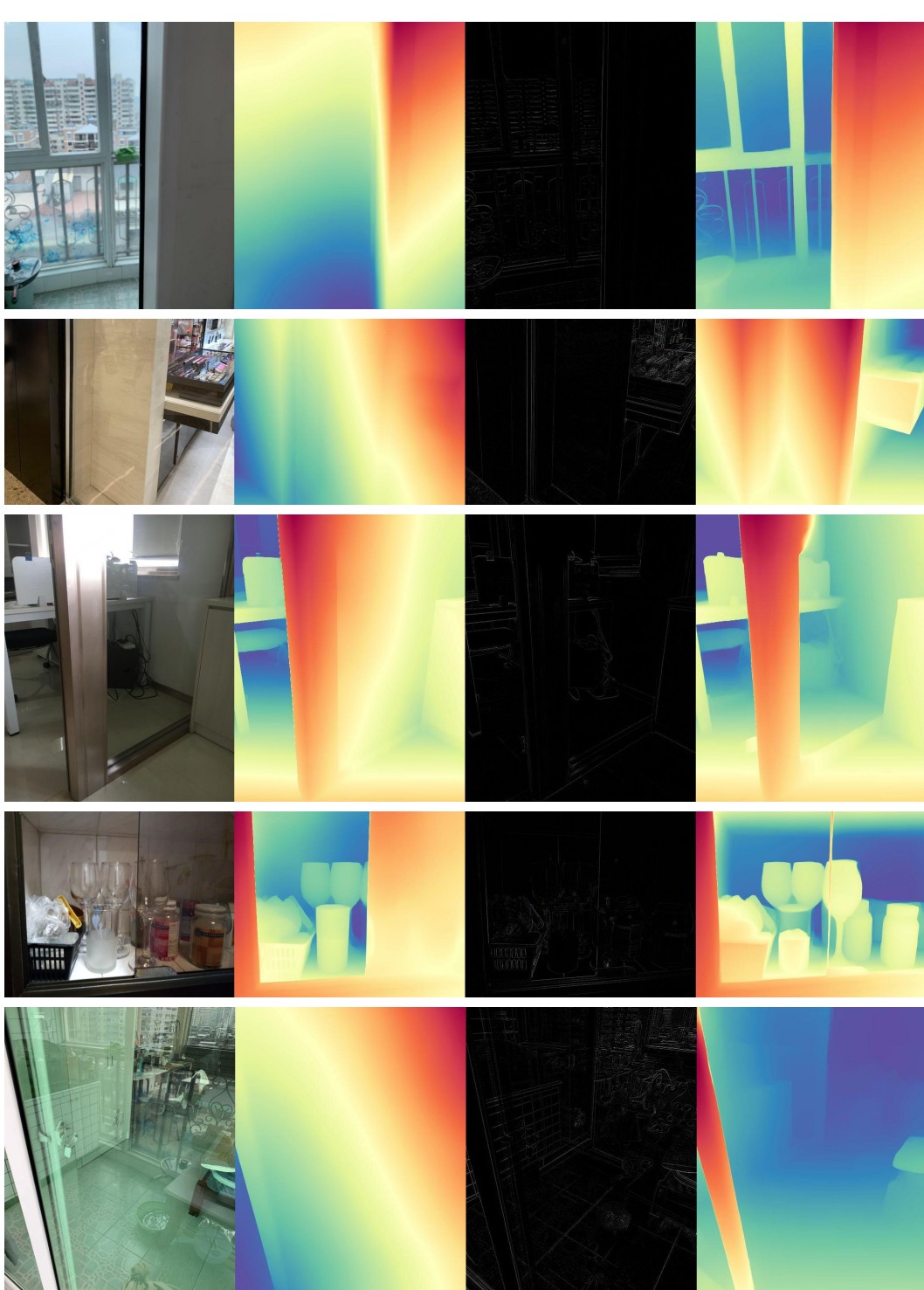

Figure F: **LVP-empowered multi-layer depth.** Each case includes an RGB image with its depth, and the Laplacian with its depth.

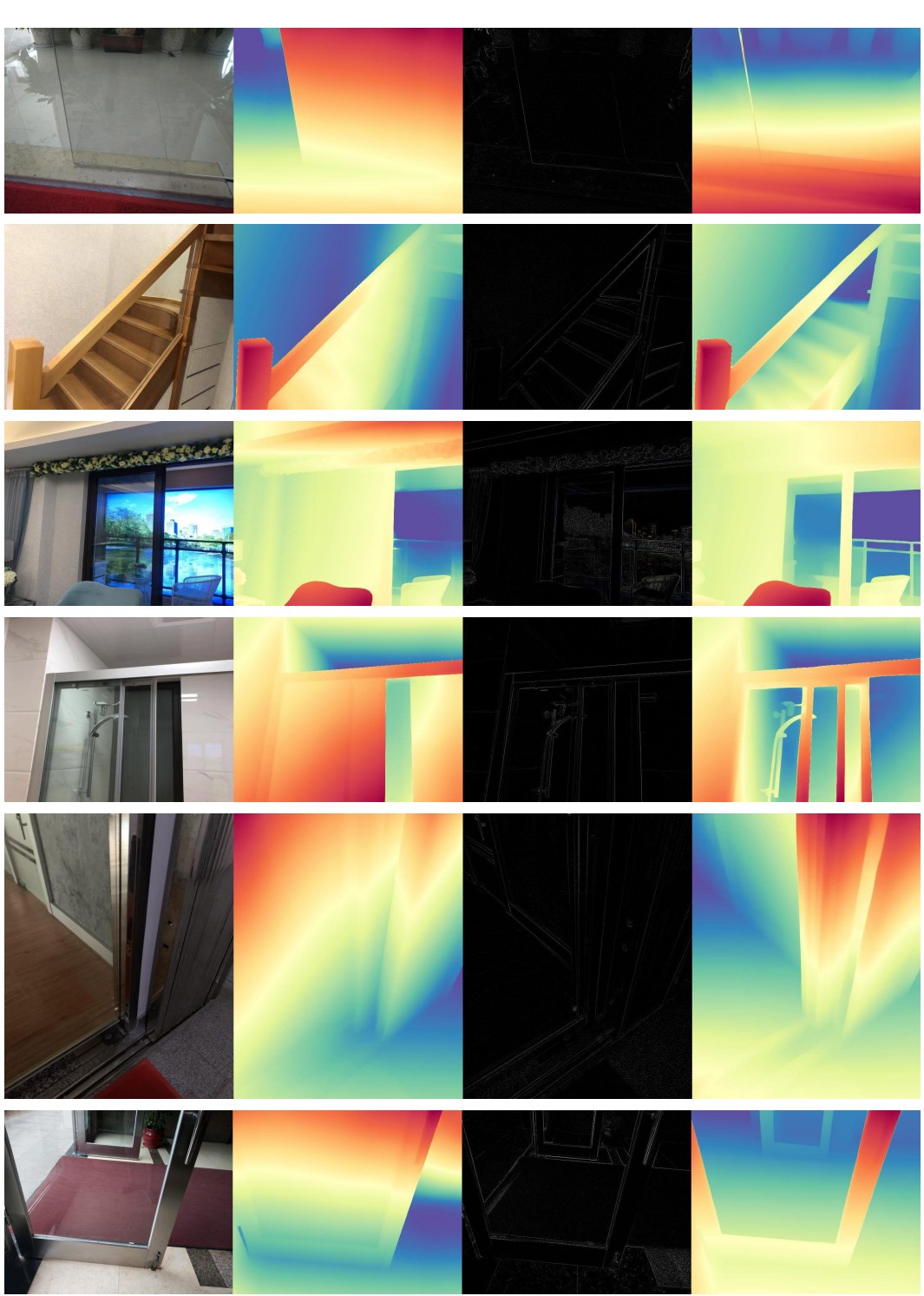

Figure G: **LVP-empowered multi-layer depth.** Each case includes an RGB image with its depth, and the Laplacian with its depth.

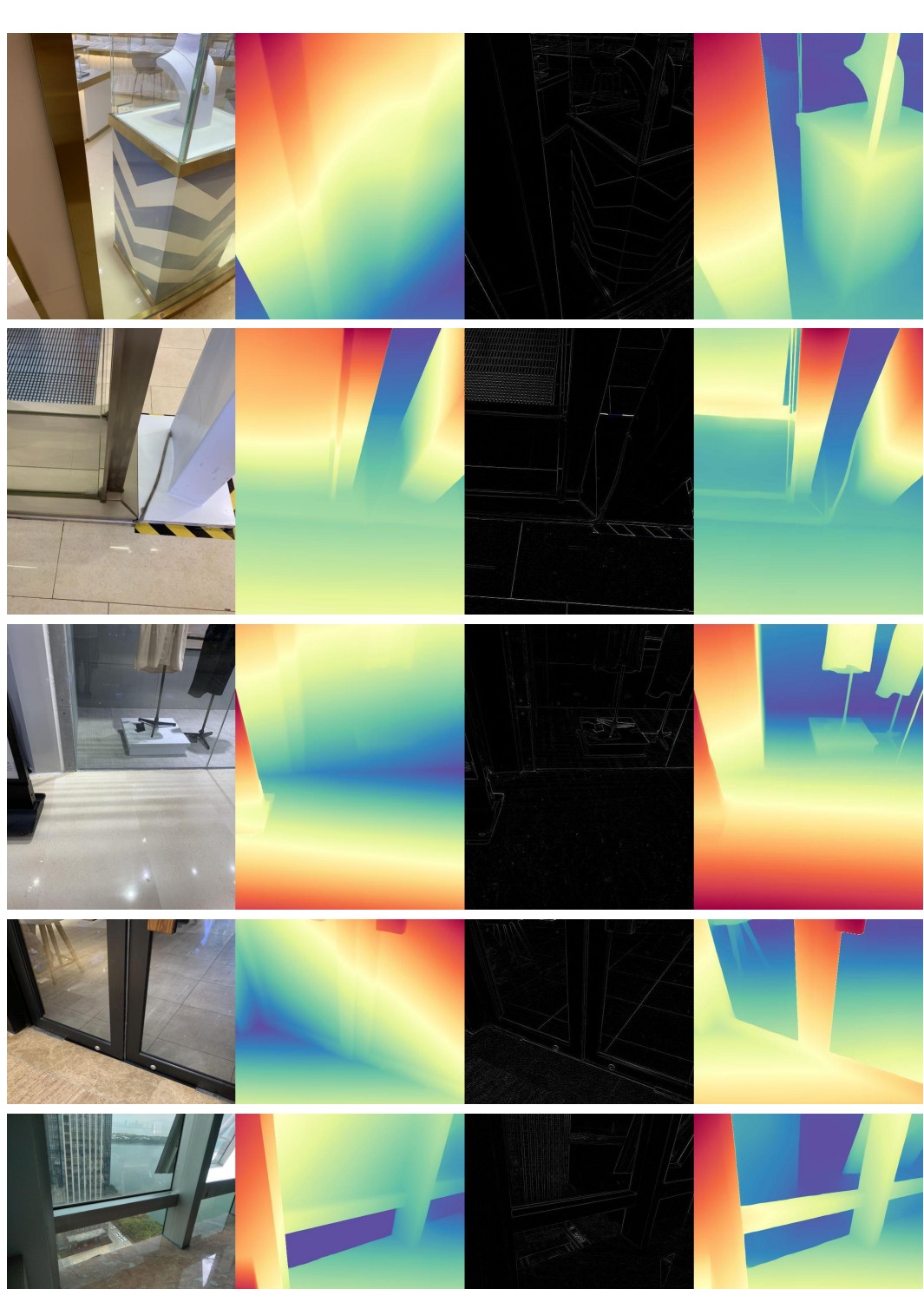

Figure H: **LVP-empowered multi-layer depth.** Each case includes an RGB image with its depth, and the Laplacian with its depth.

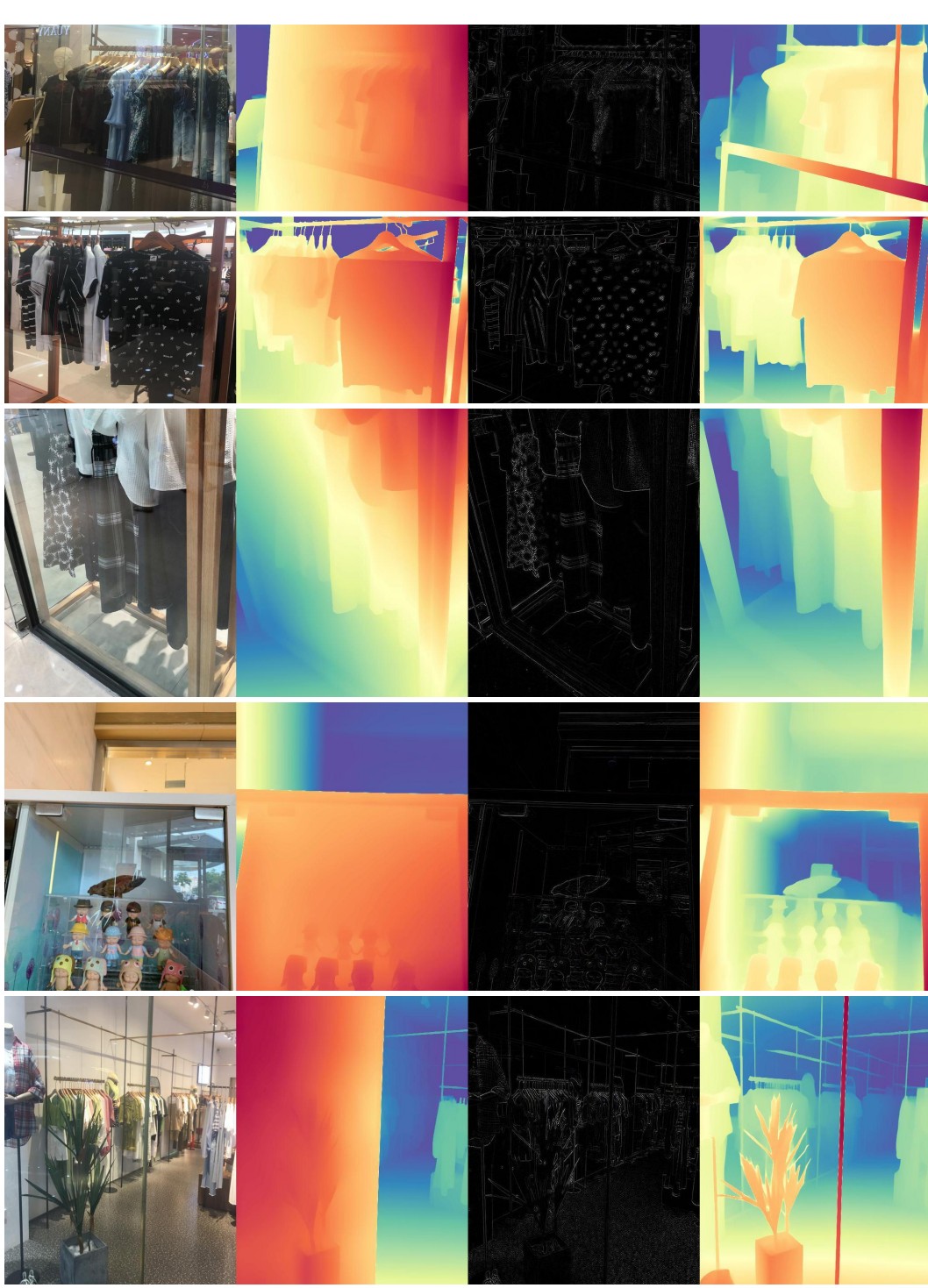

Figure I: **Failure cases** of multi-layer depth estimation via Laplacian Visual Prompting. Each case shows the RGB input, the estimated depth via RGB, the Laplacian input, and the estimated *depth* via Laplacian input.

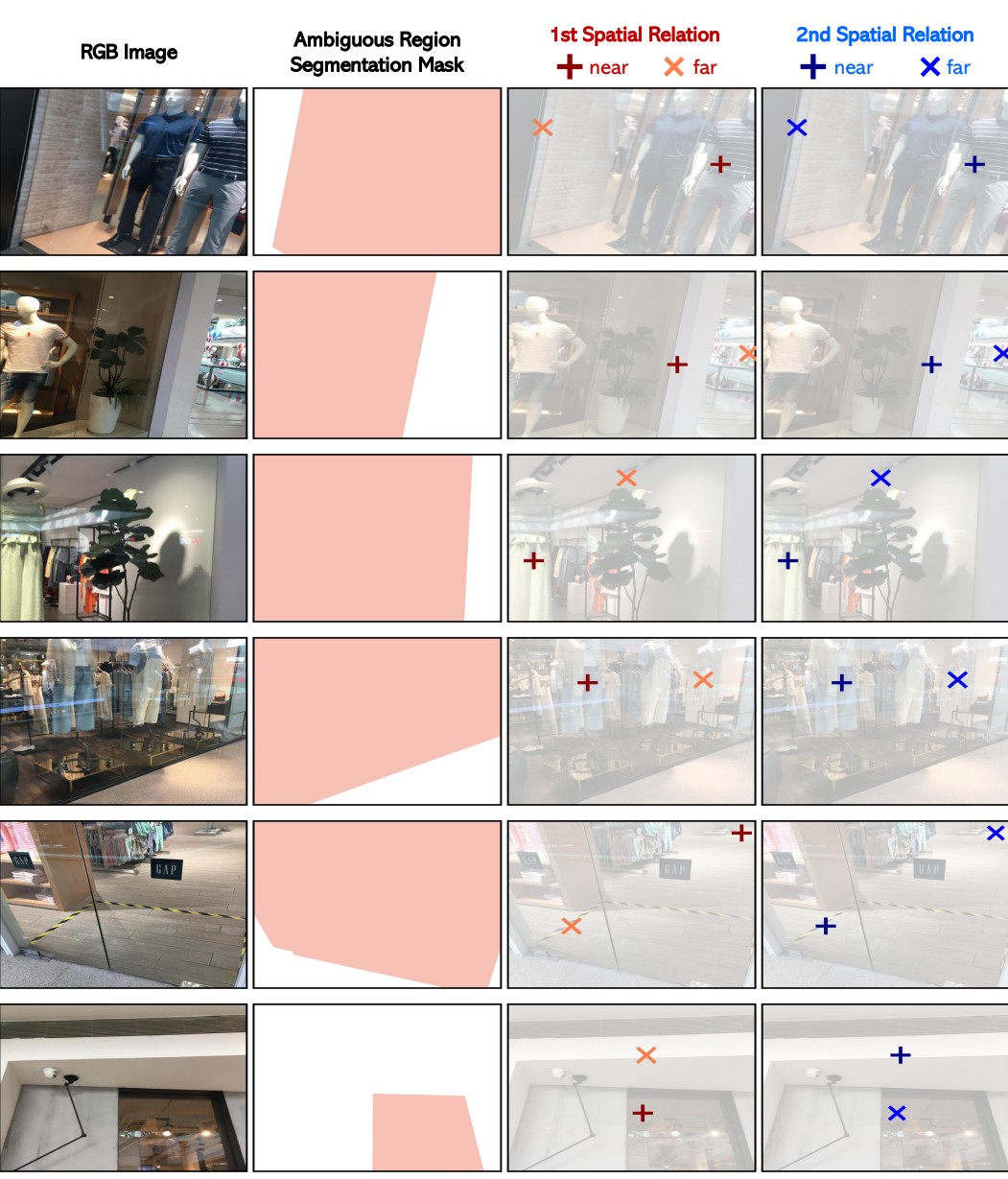

Figure J: **MD-3k benchmark for evaluating multi-layer spatial relationships.** Example images with annotated sparse point pairs are shown, illustrating ambiguous regions and relative depth relationships. The first and second spatial relation columns show ground truth annotations for near/far relationships between layers, using red and blue markers, respectively.

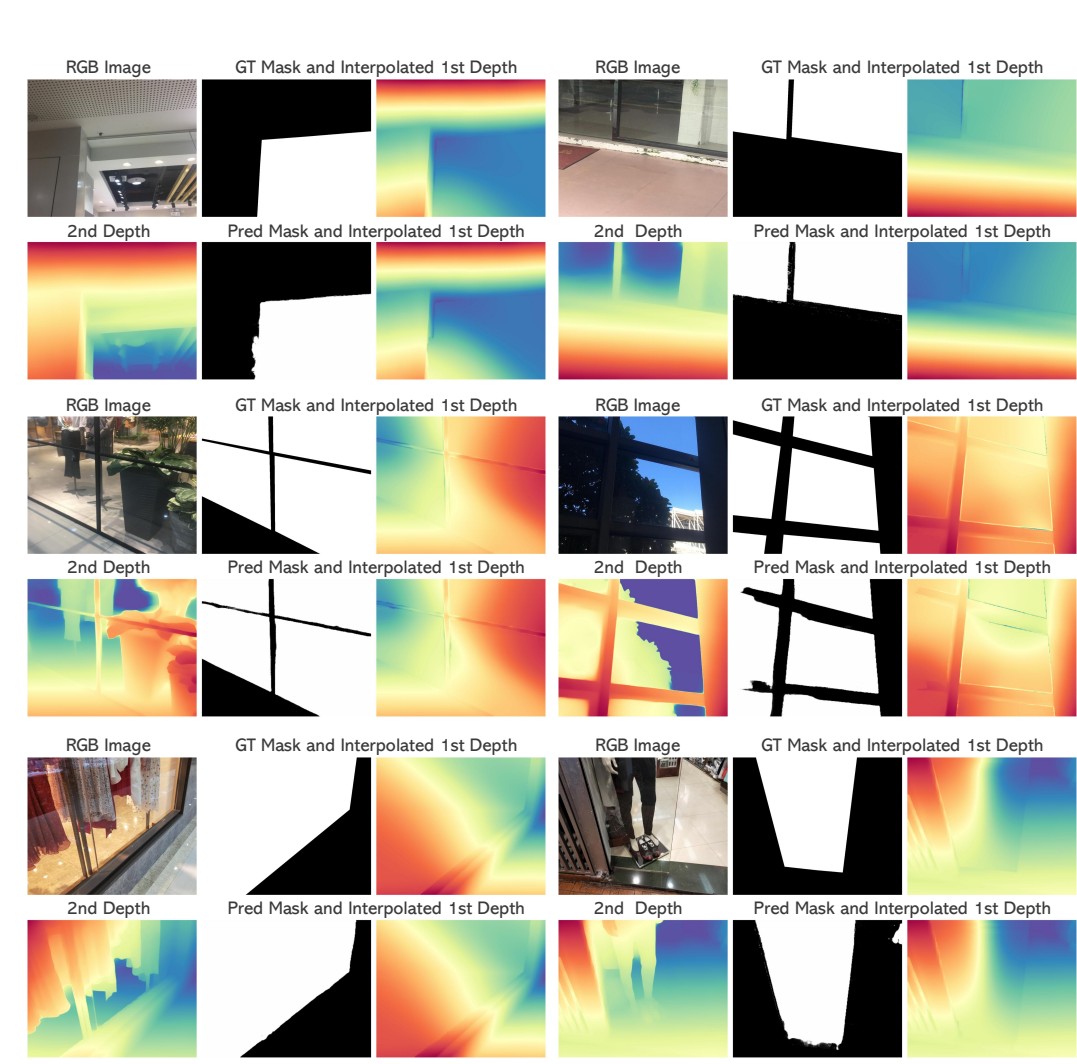

Figure K: **Multi-layer depth with extra semantic prior (successful cases).**

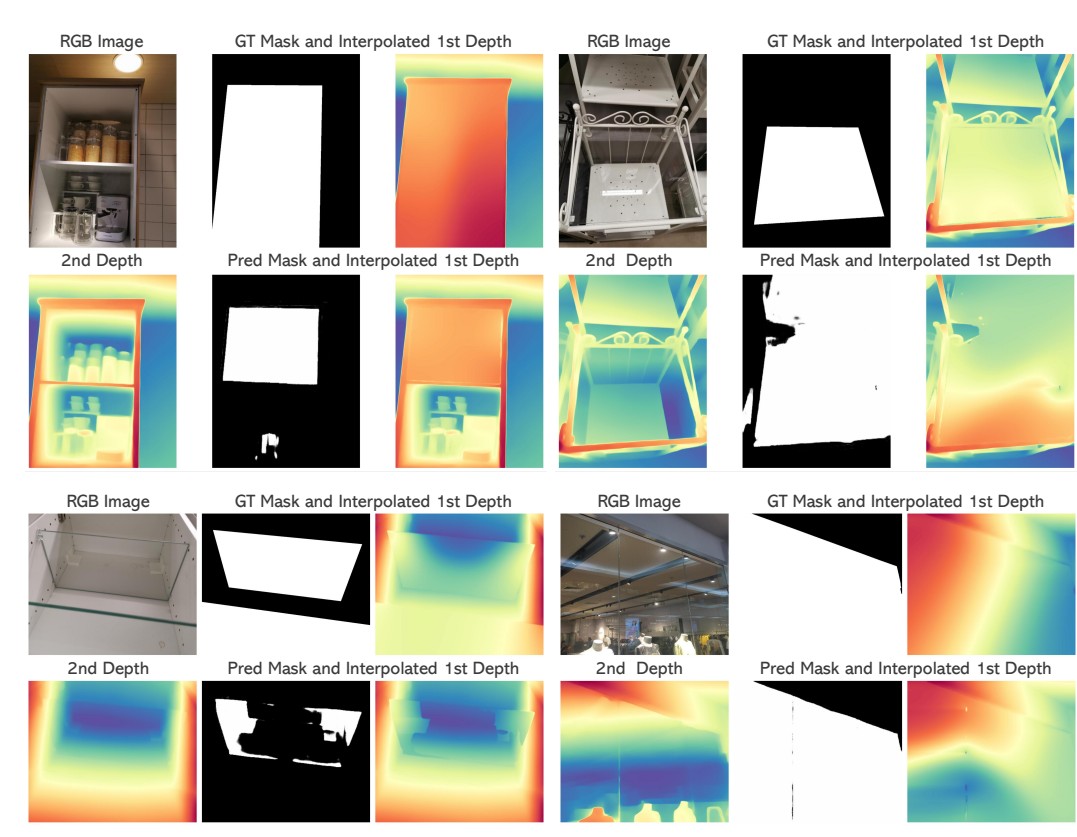

Figure L: **Multi-layer depth with extra semantic prior (failure cases).**

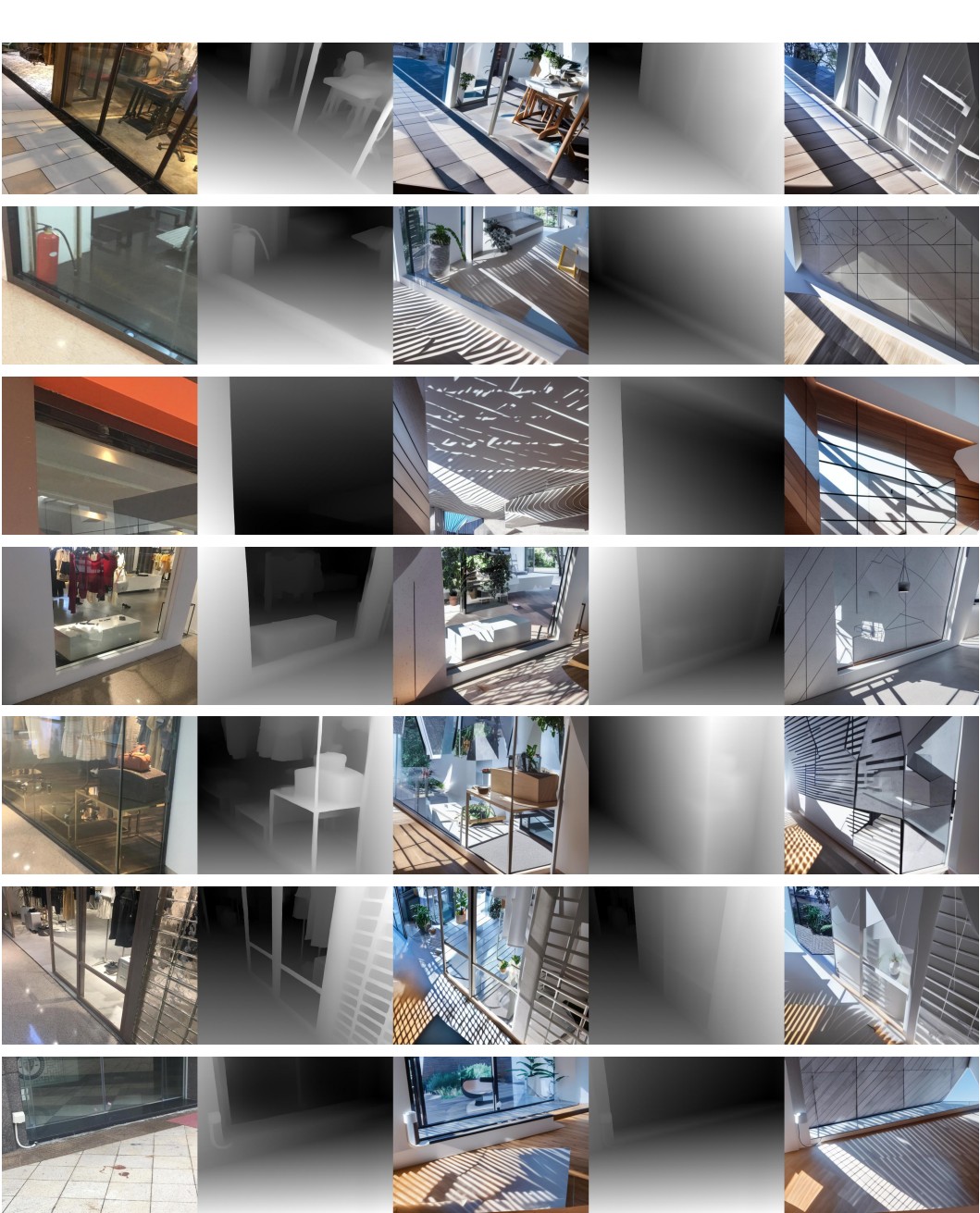

Figure M: **Multi-hypothesis spatial understanding enhances flexible geometry-conditioned visual generation.** From left to right: original RGB image, depth from Laplacian Visual Prompting with its corresponding generated RGB image, and depth from the original RGB image with its generated RGB counterpart.

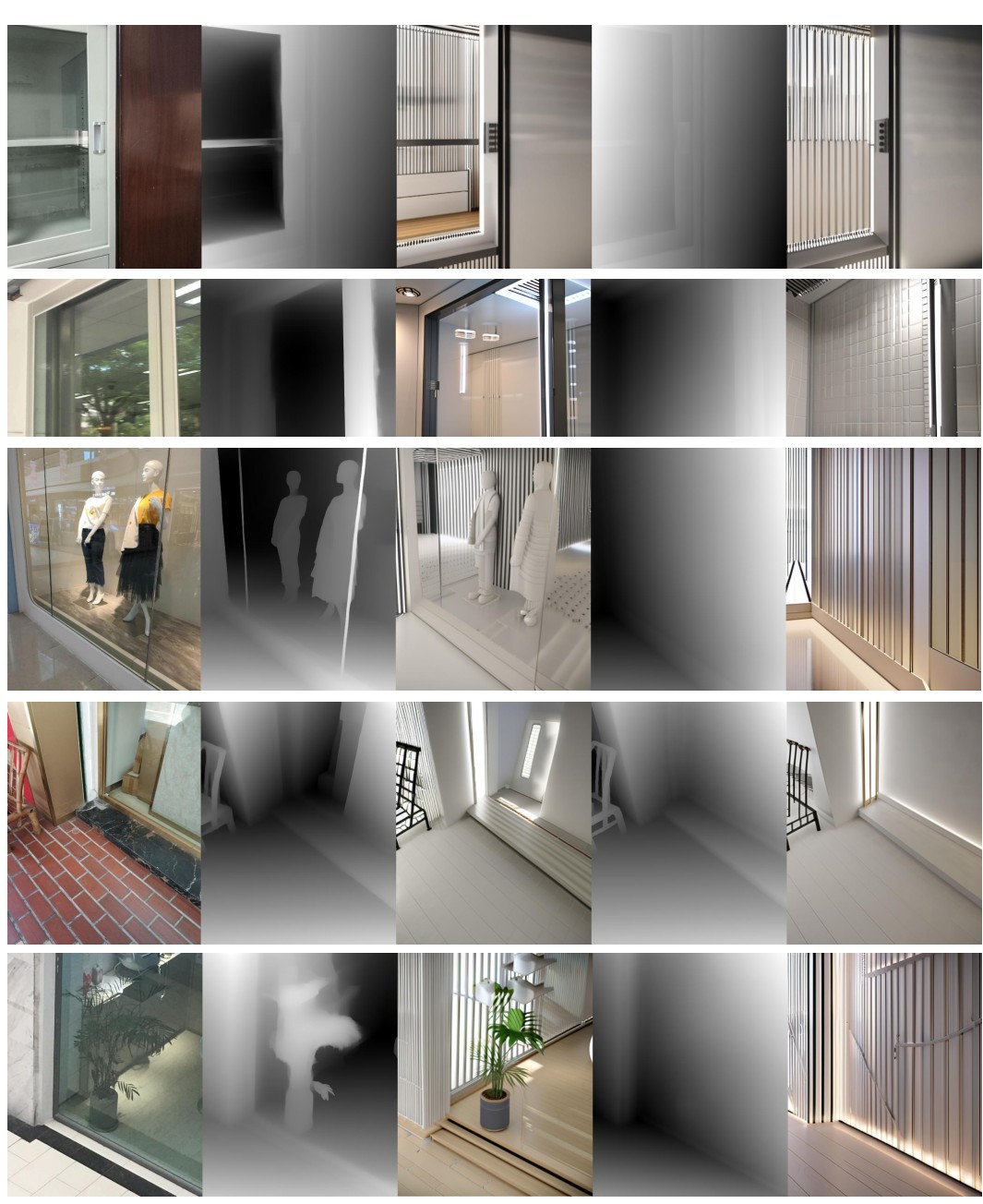

Figure N: **Multi-hypothesis spatial understanding enhances flexible geometry-conditioned visual generation.** From left to right: original RGB image, depth from Laplacian Visual Prompting with its corresponding generated RGB image, and depth from the original RGB image with its generated RGB counterpart.

