# OpenReview forum: "Can Monocular Single-Depth Foundation Models Generate Multi-Depth Hypotheses?"
_ICLR.cc/2026/Conference — ICLR 2026 Conference Desk Rejected Submission_

### Official Review · Reviewer_YVLM · 2025-10-26

**Soundness:** 2
**Presentation:** 3
**Contribution:** 2
**Rating:** 4
**Confidence:** 4

**Summary:**

In this paper, the authors study how to enable monocular depth foundation models to understand multi-layered 3D scenes with transparent glasses. Without finetuning monocular depth models, authors find that simply prompting a pre-trained model with a Laplacian of the input image, i.e., Laplacian Vision Prompting, can improve the background depth estimation. Alongside with the paper, a new benchmark named MD-3k is introduced for evaluation.

**Strengths:**

1. This paper explores how to understand multi-modal spatial relationship in the scenes with transparent glasses, which is challenging and meaningful for the community.

2. This paper introduces MD-3k, a new benchmark to systematically evaluate the spatial relationships (near vs. far) of multi-layer depth estimation.

3. Without finetuning any models, Laplacian Visual Prompting is a plug-in module for different models. It is simple and proven effective on the MD-3k benchmark.

**Weaknesses:**

1. The evaluation metrics only evaluate whether the orders of depth is correct. More depth map metrics, such as relative depth difference, should be used to show the accuracy of depth map.

2. It is not guaranteed that pre-trained monocular depth models always output foreground depth, e.g. depth of transparent glass. Therefore, the claim that ``biased single-layer estimate from vanilla monocular depth model and prediction with LVP-transformed input are complementary ‘’ may not be correct. Both of them may output the background depth. This is also illustrated in Fig. 6.

3. Using edge information, estimated with Laplacian operator, only will cause ambiguity in depth prediction, resulting in low accuracy. For example, if a checkboard (black & white) is placed on a table, the depth map from Laplacian visual prompting should not be flat since there is rich edge information on the checkboard. This problem also appears in the visualization of Fig. 10. For the fourth example, there is a dark-gray carpet on the light-gray floor, resulting in an edge in Laplacian response. As shown in 2nd depth image, there exists depth inconsistency between carpet and floor. In addition, Laplacian operator is a high-pass filter, filtering out lots of information. However, to accurately estimate depth maps, intensity information is very important, e.g. shadow or slight intensity change in a local window.

**Questions:**

1. Instead of Multi-Layer Monocular Depth Estimation, I think two-layer MDE would be more accurate because the paper only discusses the scene with two depth layers (transparent glass + objects in the back). The current pipeline may not support more depth layers, e.g. 3.

2. Would the pipeline work for curved transparent surfaces?

3. L118: Diffusion-based monocular depth models can also output multi-modal depth maps by changing the random seeds of input Gaussian noise. This should be clarified.

4. L314: Could authors explain the reason why LVP reduces 2nd layer depth preference for some models. Intuitively, LVP provides information for second layer depth and should increase 2nd layer depth preference, e.g. DA v2-L.

---

> ### Author Response · Authors · 2025-11-13
> **Reply to Reviewer YVLM**
>
> Dear Reviewer YVLM,
>
> Thank you for your feedback and for highlighting the challenging and meaningful nature of this problem. We have revised the paper to address your concerns, particularly regarding failure cases and model mechanisms.
>
> ---
>
> > **W.1. Why ordinal-only evaluation**
>
> We agree that metric evaluation would be valuable. However, this is a fundamental challenge in this problem space. Physical sensors (like LiDAR) fail to capture accurate metric depth for transparent or multi-layer scenes (as shown in **Appendix Fig. B**). This is precisely why we created **MD-3k**, the first benchmark to our knowledge, which relies on *relative ordinal* relationships. We have now added a discussion of SOTA metric depth models as a key area for future work in **Sec. 5**.
>
> ---
>
>
> > **W.2. Bias & complementary claim**
>
> Sorry for the confusion! You are right. The claim is only true when the model's default bias and its LVP-prompted bias are *different*. As **Fig. 6** shows, models like DAv1 are biased to L2 for *both* RGB and LVP inputs. In this case, our method fails to decouple the layers. This supports our re-clarified "mode-switching" hypothesis (Sec. 3.5 & 4.1): LVP is not a magic "background-finder," but a probe that *attempts* to switch modes. If the model's modes are not well-separated (as in DAv1), the perturbation fails to induce a switch.
>
> ---
>
> > **W.3. Failure cases (Checkerboard / Carpet) and edge information**
>
> This is an excellent point! The "dark-gray carpet" example (Fig. 10) is a high-frequency *foreground* texture. LVP, as a high-pass filter, would amplify this signal, and the model would likely (and incorrectly) associate it with the background. We have now explicitly added this as a failure case in our new **"Open questions" section (Sec. 5)** and **Fig. 12b**.
>
>
>
> Our goal is not to present LVP as a perfect solution, but as a *simple probe* to demonstrate that the *potential* for multi-layer understanding exists. We agree with your insight that "intensity information is very important," which is precisely why LVP outperforms Canny edge prompts (as shown in **Table 4**). The Laplacian preserves richer, *non-binary* derivative and intensity-related information, providing a more robust prompt than binary edges alone.
>
> ---
>
>
>
> > **Q.1. Two-Layer is more specific scope**
>
> Thank you for the suggestions! Our method and benchmark focus specifically on the two-layer case. We have revised the paper (e.g., in **Sec. 1, 3.1, and 5**) to be more precise with this terminology.
>
> ---
>
> > **Q.2. Can LVP handle curved surfaces?**
>
> Great question! The answer is yes! LVP operates on the *image frequency*, not the 3D geometry. As long as the background seen *through* the curved glass has high-frequency details, LVP could amplify them. We have now added **Fig. 12a**, which explicitly shows several cases where LVP successfully handling curved transparent glass surfaces.
>
> ---
>
> > **Q.3. Diffusion models as alternative**
>
> This is a great point! We have added  clarification and discussion to our **Related Work (Sec. 2)**. While diffusion models can theoretically produce multi-modal outputs via different noise seeds, this is (1) **non-deterministic** and (2) **very slow** (requiring multiple full sampling runs). (To the best of our knowledge, there is no work exploring diffusion models' capabilities to tackle multi-layer depth prediction for ambiguous scenes.) Our LVP is (1) **deterministic** (one LVP input gives one complementary output) and (2) **very fast** (one cheap `conv` operation + one forward pass).
>
> ---
>
> > **Q.4. LVP reducing 2nd layer preference**
>
> To clarify, the 'mode-switching' hypothesis (re-clarified in Sec. 3.5) is model-dependent. For architectures like DPT or ZoeDepth, the intrinsic bias is already toward the 2nd layer (background), as shown in Fig. 6 and also Table A in the Appendix. In these cases, LVP acts as a disruptive prompt: it destabilizes this default preference (Layer 2) and forces the model to converge on the alternative stable mode (Layer 1). Therefore, a reduction in SRA(2) effectively signals a successful switch to the foreground. While initially counter-intuitive, this confirms LVP’s ability to expose latent interpretations that the model ignores by default.
>
> ---
>
> We would like to thank you again for your helpful feedback. All of our revisions in the paper are **highlighted in blue** for your convenience. We are **happy to receive further comments and address any remaining concerns.**
>
> If you feel our response and revisions have adequately addressed your concerns, **we would be grateful if you would consider your rating.** Thank you!

---

> ### Author Response · Authors · 2025-11-24
> **Looking Forward to Discussion with You**
>
> Dear Reviewer YVLM,
>
> We sincerely appreciate your time and effort in reviewing our manuscript and providing valuable feedback.
>
> In response to your insightful comments regarding failure cases and scope, we have made the following revisions to our manuscript:
>
> * **Curved Surfaces:** We have added **Fig. 12a**, explicitly demonstrating that LVP successfully generalizes to curved transparent surfaces, handling non-planar geometry.
> * **Failure Cases (High-Freq Foreground):** We addressed your concern regarding textured foregrounds (e.g., carpets/checkerboards) by adding a detailed **"Open questions" (Sec. 5)** and **Fig. 12b**, acknowledging that LVP may fail when foregrounds contain high-frequency patterns.
> * **Clarified Terminology:** As suggested, we have revised the text throughout (e.g., **Sec. 1, 3.1, 5**) to be more precise, referring to our method as **"two-layer"** where appropriate.
> * **Edge vs. LVP:** We clarified the comparison in **Sec. 4.5**, highlighting that LVP preserves intensity-rich information compared to binary Canny edges (Table 4).
> * **Diffusion vs. LVP:** We added a clarification in **Sec. 2** distinguishing our deterministic, fast LVP approach from the non-deterministic, slow sampling of diffusion models.
>
> We hope these revisions adequately address your concerns.
>
> We look forward to actively participating in the Author-Reviewer Discussion session and welcome any additional feedback you might have on the manuscript or the changes we have made.
>
> Once again, we sincerely thank you for your contributions to improving the quality of this work.
>
> Best regards,
>
> The Authors of Submission 1835

---

> ### Author Response · Authors · 2025-11-27
>
> Dear Reviewer,
>
> As the discussion phase is ending, we wanted to ensure that our previous response and revisions have adequately answered your questions.
>
> Your insights have helped us significantly improve the paper. If you are satisfied with the changes, we would appreciate it if you would consider updating your rating. Please let us know if there are any final points you'd like us to address or discuss. Thank you!
>
> Best regards,
> Authors of Paper 1835

---

### Official Review · Reviewer_m2ax · 2025-10-30

**Soundness:** 2
**Presentation:** 3
**Contribution:** 3
**Rating:** 4
**Confidence:** 5

**Summary:**

This paper presents a training-free approach for generating a second, alternative depth layer for a single input image by feeding to the monocular depth model at hand the Laplacian-filtered version of the image. Moreover, the authors curate the new MD-3k benchmark for two-layer depth estimation which is sourced from the existing GDD dataset, by annotating it with pairwise depth ordering labels across two different depth layers, and introduce new metrics to evaluate two-layer depth predictions. Comparisons and ablations on the new benchmark analyze the depth-layer biases of several central monocular relative depth estimators and show that the proposed input-space visual prompting can yield decent two-layer depth predictions.

**Strengths:**

1. The examined problem of two-layer monocular depth estimation is challenging, interesting and of practical use to computer vision practitioners.

2. The core idea of input-space visual prompting to allow a training-free second depth layer prediction is clever and well-motivated.

3. The compilation of a new benchmark to quantitatively evaluate methods' performance on the newly considered task is an important contribution, besides the methodological aspect of the paper.

**Weaknesses:**

1. Soundness of central hypothesis. The core hypothesis of the LVP-based method that amplifying high frequencies will prioritize the prediction of the second, further depth layer with strong textures breaks in the cases where the depth model at hand is already biased towards the second depth layer. This is the case for models such as DAv1 and DAv2-O (L. 426), for which the employed LVP-based two-layer estimation unintuitively deteriorates performance as model size is increased. My perception is that the proposed method works well only when the depth estimator at hand is indeed biased to originally predict the first, low-frequency depth layer, and fails otherwise. I would expect that ideally such a training-free method should first determine which bias the estimator has and adaptively construct the visual prompt either as a low-pass or as a high-pass signal for obtaining the second depth layer accordingly. Table 4 is not fully helpful in examining this setting, as it only shows the "Reverse" subset comparison of low-pass vs. high-pass prompts, while omitting the "Same" subset, which might reveal an opposite picture to the former subset. The soundness of the high-pass second-layer hypothesis is also challenged by the finding (L. 344-345) that DPT, ZoeDepth, and DAv2-SB flip their predictions from the second layer to the first when prompted with the Laplacian, which is against the provided intuition.

2. Performance trails that of easy baselines. While the proposed training-free method exhibits decent performance for two-layer prediction, Table 2 reveals that the simple baseline of prompting the depth model with semantic predictions results in even better performance. Note here that the boldface notation in this table is inconsistent, highlighting the results of the proposed method although for "Overall" and "Reverse" it is another method that scores the highest. Given that the current state of the art in segmentation is quite advanced, with models such as SAM delivering accurate masks for diverse monocular inputs which capture well edges that correspond to depth gradients, the utility and relevance of the proposed Laplacian filtering, which performs below-par to segmentation-based prompts, is questioned. This comparison also contains a vague aspect: the authors mention that "extra semantic priors" are used as prompts. How are the Laplacian and the semantic prompts combined, if they are? If they are, what is the comparison of using semantic priors rather as drop-in replacement of Laplacian prompts - which prompt alone is better?

3. Quantitative comparison of single-layer vs. multi-layer methods and respective baselines. The proposed evaluation can accurately capture the comparative performance of different two-layer methods. However, I do not find how the proposed framework can actually tell whether a two-layer prediction does not introduce more errors compared to a single-layer prediction than the correctly predictions it makes for alternative-layer pixels. I also understand that the current evaluation is restricted to the ground-truth masks that indicate the presence of two layers in the respective image regions. What about the consistency of the two different predictions in single-layer regions of the image? Moreover, it is not clear to me what the performance of a dummy, e.g. random baseline would be in the setting of the main comparison in Table 1. Such a measure could serve as reference to better interpret the "Overall" and "Same" figures in that table. In my understanding, simply copying the primary depth prediction to the secondary one would yield a score of 100% for "Same" and 0% for "Reverse". I would like to thus see what the respective "Overall" score would be for that baseline, in order to better understand how much the presented models improve the "Overall" score comparatively.

4. Ordinal-only evaluation. While the presented benchmark is useful for evaluating the ordinal correctness of predicted multi-layer depth maps, it ignores the continuously-valued accuracy of the predictions, with measures such as $\delta_1$ etc. Moreover, only the relative depth estimation scenario is experimentally examined. Foundational monocular metric depth estimators such as UniDepth [A], UniK3D [B], UniDepthV2 [C], Metric3D v2 [D], and Depth Pro [E] are ignored, even though both their outputs are compatible with the relative depth estimation setting which is examined and they could serve as baselines for metric two-layer depth estimation experiments using the proposed LVP approach.

[A] Piccinelli et al.: UniDepth: Universal monocular metric depth estimation. In CVPR, 2024.

[B] Piccinelli et al.: UniK3D: Universal camera monocular 3D estimation. In CVPR, 2025.

[C] Piccinelli et al.: UniDepthV2: Universal monocular metric depth estimation made simpler. IEEE T-PAMI, 2025 (to appear).

[D] Hu et al.: Metric3D v2: A Versatile Monocular Geometric Foundation Model for Zero-shot Metric Depth and Surface Normal Estimation. IEEE T-PAMI, 2024.

[E] Bochkovskii et al.: Depth Pro: Sharp monocular metric depth in less than a second. In ICLR, 2025.

**Questions:**

Cf. my questions in the Weaknesses section.

---

> ### Author Response · Authors · 2025-11-13
> **Reply to Reviewer m2ax**
>
> Dear Reviewer m2ax,
>
> We thank you for your "absolutely certain" assessment and detailed, constructive suggestions. Your feedback has led to significant improvements in our paper, particularly in the analysis of our hypothesis and baselines.
>
>
> ---
>
>
> > **W.1. Clarification of central hypothesis (LVP breaks, scaling laws)**
>
> Sorry for the confusions! Our initial "high-pass = background" hypothesis was indeed not clear. We have revised the paper (notably in **Sec. 3.5, 4.1, and 4.3**) to introduce a more robust **"mode-switching" hypothesis** statement to clarify the phenomenon that some depth foundation models can exhibit a complementary depth prediction under the Laplacian-transformed RGB image input compared to the depth under the standard RGB input:
> - Models have a *default bias* (e.g., DAv2 $\to$ L1, DPT $\to$ L2), as seen in **Fig. 6**, which can be examined via our MD-3k benchmark.
> 2.  LVP acts as a perturbation that "knocks" some depth foundation models into the *complementary* stable mode (**Sec. 3.5**). For example, DAv2 (L1 default) flips to L2 while DPT (L2 default) flips to L1.
> -  This *also* explains the scaling effect (**Fig. 8a** and **Sec. 4.3**):
>     * If RGB/LVP have *opposite* biases (DAv2, DAv2-I), scaling *improves* ML-SRA because the larger model better separates the two modes.
>     * If RGB/LVP have the *same* bias (DAv1, DAv2-O), scaling *hurts* ML-SRA because the model becomes *more confident* in its single default mode, making it *harder* for LVP to switch.
> -  As requested, we have added the **"Same" subset results for the Gaussian (low-pass) prompt to Table 5**. It confirms that a low-pass filter *fails* to switch modes on the "Reverse" set, supporting our high-pass (perturbation) hypothesis.
>
> ---
>
> > **W.2. Performance trails "easy" baselines (Table 2)**
>
> We apologize for the lack of clarity and we really appreciate your valuable suggestion! We have revised **Sec. 4.2** to make this comparison explicit:
> -  Our training-free, *single-model* LVP achieves **75.5%** ML-SRA.
> -  The "semantic" baseline ("+ Predicted Mask") achieves **75.8%**. This baseline is *not* "easy." As detailed in **Appendix C.1**, it is a complex, multi-step pipeline that requires: (1) a *separate, powerful segmentation model*; (2) a specific depth model (DAv1-L) chosen for its known bias; and (3) a strong *geometric assumption* of planar surface interpolation.
> -  Our LVP method achieves a highly competitive result with **no extra models, parameters, or such geometric assumptions**. In fact, our new **Fig. 12a** shows that LVP *can* handle curved transparent surfaces, where the planar assumption would fail.
>
> ---
>
> > **W.3. Performance of a dummy baseline**
> This is an excellent suggestion. We have added this "dummy" ideal-performance baseline analysis to **Sec. 4.2** and **Table 1**:
> * An ideal baseline that copies the primary (RGB) depth would achieve 100% (upper-bound) on the "Same" subset (1,783 pairs) and 0% on the "Reverse" subset (1,378 pairs).
> * The weighted "Overall" score for this baseline is **56.4%**.
> * Our DAv2-L's score of **75.5%** represents a **+19.1% absolute improvement**, clearly demonstrating that LVP is eliciting meaningful, complementary geometric information.
>
> ---
>
> > **W.4. Ordinal-only evaluation and metric depth**
>
> This is an insightful point which we have considered.
> * **Why Ordinal?** We chose an ordinal-only evaluation (SRA) precisely because collecting dense, multi-layer *metric* ground truth is an unsolved problem, largely due to the failure of physical sensors like LiDAR in these ambiguous regions (as shown in **Appendix Fig. B**). This fundamental challenge is what motivated our work.
> * **Metric Models:** We thank you for the valuable insights and the suggested great papers!  We have included the suggested Depth Pro,  UniDepthv2, and UniK3D methods in our benchmarking on MD-3k (See **Fig. 6 and Table 1**), and included the metric-scale multi-layer depth evaluation  as a key area for future work  in our revised **Sec. 5**.  Moreover, **all** of the suggested papers are cited in our revised manuscript.
>
> We fully  agree with you that it is a valuable next step to evaluable metric-scale depth maps  from SOTA metric-scale depth foundation models. However, we still want to highlight that this is a non-trivial and challenging goal due to the grand challenge of lacking multi-layer metric-scale depth collection and annotation methods.
>
> ---
>
> We would like to thank you again for your meticulous review, which has led to a much stronger and more clearly-argued paper. All of our revisions in the paper are **highlighted in blue** for your convenience. We are **happy to receive further comments and address any remaining concerns**.
>
> If you feel our response and revisions have adequately addressed your concerns, **we would be grateful if you would consider your rating**! Thank you!

---

> ### Author Response · Authors · 2025-11-24
> **Looking Forward to Discussion with You**
>
> Dear Reviewer m2ax,
>
> We sincerely appreciate your time and effort in reviewing our manuscript and providing valuable feedback.
>
> In response to your insightful comments regarding the baselines and our central hypothesis, we have made the following revisions to our manuscript:
>
> * **Added "Dummy" Baseline:** As suggested, we added a "dummy" baseline to **Table 2**. This ideal baseline achieves 56.4% ML-SRA, highlighting that our method (+19.1% improvement) effectively elicits meaningful geometric information.
> * **Clarified Semantic Baseline:** We revised **Sec. 4.2** to clarify that the "semantic" baseline is not "easy"—it requires an extra segmentation model and strong planar assumptions. We emphasize that our single-model LVP achieves comparable performance without these extra dependencies.
> * **Validating High-Pass Hypothesis:** We added the "Same" subset results for the Gaussian (low-pass) prompt to **Table 5**. This confirms that high-pass signals are essential for switching modes in ambiguous "Reverse" cases.
> * **Refined Core Hypothesis:** We revised **Sec. 3.5, 4.1, and 4.3** to clarify the **"mode-switching" hypothesis** (instead of just "background extraction"), which better explains why scaling laws affect different models differently based on their default bias.
> * **Addressed Metric Depth:** We have cited the suggested SOTA metric-depth models (UniDepth, Metric3D, etc.) and discussed the lack of multi-layer metric ground truth as a limitation and future direction in **Sec. 5**.
>
> We hope these revisions adequately address your concerns.
>
> We look forward to actively participating in the Author-Reviewer Discussion session and welcome any additional feedback you might have on the manuscript or the changes we have made.
>
> Once again, we sincerely thank you for your contributions to improving the quality of this work.
>
> Best regards,
>
> The Authors of Submission 1835

---

> ### Author Response · Authors · 2025-11-27
>
> Dear Reviewer,
>
> As the discussion phase is ending, we wanted to ensure that our previous response and revisions have adequately answered your questions.
>
> Your insights have helped us significantly improve the paper. If you are satisfied with the changes, we would appreciate it if you would consider updating your rating. Please let us know if there are any final points you'd like us to address or discuss. Thank you!
>
> Best regards,
> Authors of Paper 1835

---

### Official Review · Reviewer_sERC · 2025-10-31

**Soundness:** 2
**Presentation:** 3
**Contribution:** 3
**Rating:** 6
**Confidence:** 3

**Summary:**

The paper shows that monocular depth foundation models, although trained to output a single deterministic depth per pixel, actually encode latent multi-layer geometry.

The authors introduce Laplacian Visual Prompting (LVP), a training-free input perturbation that feeds the model a Laplacian-filtered image. This high-frequency “prompt” suppresses smooth foreground surfaces (e.g., glass) and coaxes the frozen model to produce a complementary depth map for the background.

Evaluated on the proposed MD-3k benchmark of ambiguous, multi-layer scenes, LVP separates foreground and background, turning an off-the-shelf depth model into a multi-hypothesis estimator without any retraining or extra parameters.

**Strengths:**

Reveals a previously unreported property: single-depth foundation models implicitly encode multi-layer geometry. This reveal that the geometric capacity of depth foundation models is richer than their standard outputs suggest, and open a new path toward probing and harnessing hidden representations for more complete 3D understanding.

Introduces Laplacian Visual Prompting (LVP)—a simple, training-free input transform that converts any frozen depth model into a multi-hypothesis estimator. The idea of using a classical Laplacian filter as a “visual prompt” is elegant and novel.

Provides MD-3k, the dedicated benchmark for transparent / multi-layer depth evaluation.

LVP is tested across several foundation models and consistently produces complementary depth maps.

**Weaknesses:**

The authors argue that the Laplacian prompt suppresses low-frequency foreground glass signals and highlights high-frequency background details (from the first layer to the second). However, Fig. 6 shows that for some models (DPT, ZoeDepth, DAV2-SB) the behavior flips: the Laplacian prompt draws the prediction toward the foreground glass rather than the background (from the second layer to the first). The paper does not explain why a single high-pass perturbation drives different models toward opposite posterior modes, nor how practitioners can anticipate which layer a given model will favor.

The manuscript did not discuss several studies that likewise expose hidden capabilities of pretrained vision models, such as “Emergent Correspondence from Image Diffusion,” “Cross-View Completion Models are Zero-shot Correspondence Estimators,” “Easi3R: Estimating Disentangled Motion from DUSt3R Without Training,” and “Video Models are Zero-shot Learners and Reasoners.” A clearer comparison would help position LVP’s novelty and clarify how it complements these zero-shot 3D reasoning approaches.

**Questions:**

Why does a Laplacian-filtered image make sense to the model? The prompt is clearly OOD: the network was never trained on these high-pass inputs.

Inconsistent layer preference across models (Fig. 6). For DPT, ZoeDepth, and DAV2-SB, the Laplacian prompt highlights the foreground instead of the background. Can you give practitioners a simple heuristic for predicting whether a given model will swap layers or not?

Impact on the original (first-layer) prediction. When LVP is applied, the original layer is degraded. In applications, one may need both layers: how should we decide where and when to apply LVP? Please quantify the accuracy drop on the first layer and discuss possible mitigations such as spatially selective prompting or blending strategies.

Can the method be used to automatically combine two layers prediction?

---

> ### Author Response · Authors · 2025-11-13
> **Reply to Reviewer sERC**
>
> Dear Reviewer sERC,
>
> Thank you for your valuable feedback. We are encouraged that you found our core finding "previously unreported" and the LVP idea "elegant and novel".
>
> We have made significant revisions, particularly to our theoretical justification in **Sec. 3.5**, based on your key insights.
>
>
> ---
>
>
> > **W.1 & Q.2. Inconsistent layer preference (Fig. 6) and the practitioner's heuristic**
>
> This is a critical observation, and your question really helped us clarify our hypothesis. We now clarify in our revised **Sec. 3.5 (Theoretical Analysis)** and **Sec. 4.1**. A more accurate hypothesis that explains the phenomenon we have observed  for several depth foundation models is that LVP acts as a *mode-switching* operator of depth layer prediction bias.
>
> -  **Default Bias:** Each frozen model has a *default bias* (its primary posterior mode), which can be probed via our MD-3k benchmark. As **Fig. 6** shows, for DAv2, this is Layer 1 (glass). For DPT, this is Layer 2 (background).
> -  **Mode-Switching:** LVP is a strong, OOD perturbation for some depth foundation models. As we now theorize in **Sec. 3.5**, this "knocks" the model out of its default, stable prediction and forces it to settle into the *next available stable mode*, i.e., the complementary hypothesis.
>
> This single "mode-switching" hypothesis explains the behavior exhibited by several depth foundation models: DAv2 (L1 default) flips to L2, while DPT (L2 default) flips to L1. The heuristic for a practitioner is:
> **(1) Run the standard RGB input to find the model's default bias (using our $\alpha$ metric in Sec. 3.3). (2) LVP will then be used to prob the *other* hypothesis.**
>
> ---
>
> > **W.2. Missing related work on Emergent Properties**
>
> Thank you for these excellent references and valuable suggestions! You are precisely correct that our work is correlated to this important line of research on emerging properties. Our finding that some models trained for single-depth output *innately* encode multi-layer geometry is a direct contribution to this field. We have now added these citations and a discussion to our **Introduction (Sec. 1)** to situate our work in this context, framing it as a new form of *probing* for emergent 3D understanding. In addition, we have added a comprehensive new subsection, **Sec. E.3** (Emerging Properties for Visual Models), in the appendix. This section provides a detailed review of how generative and geometric pretraining strategies yield training-free correspondences and latent 3D structures, further contextualizing LVP within this broader landscape.
>
>
> ---
>
>
> > **Q.1. Why does an OOD (Laplacian) input make sense to the model?**
>
> This is a great and fundamental question. Our answer, now clarified in **Sec. 3.5** and **Sec. 5**, is threefold:
> -    We view this as an *emergent property* for certain models. The fact that the model *doesn't* fail catastrophically, but instead produces a *coherent, alternative* geometric structure, is the key discovery.
> -   As the model was trained on high-frequency signals (e.g., edges) present in all-natural RGB image, LVP simply *isolates* this structural signal. The model generalizes mainly because (we believe) the high-frequency components are the fundamental factor to distinguish depth in 3D space.
> -   As we now show in our new **Fig. 12** and discuss in **Sec. 5**, this OOD robustness is even broader than just LVP; models can produce meaningful depth even from *other* high-frequency prompts like edge maps and semantic masks. Also, on standard depth dataset DA-2k, the performance under LVP will not degrade a lot compared to RGB input (see Table A).
>
> ---
>
> > **Q.3. & Q.4. Impact on original depth prediction / Combining layers**
>
> Thank you for this practical question. We have added clarifications to **Sec. 4.4** and **Sec. 5** based on this.
> *  On the standard non-ambiguous dataset DA-2k (see **Table A**), the depth prediction under LVP will not degrade a lot compared to original RGB input. Intriguingly, we observe generalization of diverse models under LVP for non-ambiguous scenes (thought this is not the main focus of this paper).
> * In a practical application, LVP is not meant to *replace* the RGB input; it's meant to *complement* it. If the depth maps predicted from LVP and RGB inputs differ a lot, then they can be viewed as two layers of depth for further fusion, which we leave for future work. As suggested, we have included the blending of multiple depth layers as a valuable next step in our **"Open questions" (Sec. 5)**.
>
> ---
>
> We would like to thank you again for your very insightful and valuable suggestions, which have significantly strengthened our paper's core hypothesis. All of our revisions in the paper are **highlighted in blue** for your convenience. We are **happy to receive further comments and address any remaining concerns**.
>
> If you feel our response and revisions improve the quality of our work, **we would be grateful if you would consider your rating**. Thank you!

---

> ### Author Response · Authors · 2025-11-24
> **Looking Forward to Discussion with You**
>
> Dear Reviewer sERC,
>
> We sincerely appreciate your time and effort in reviewing our manuscript and providing valuable feedback.
>
> In response to your insightful comments regarding the model behaviors and theoretical grounding, we have made the following revisions to our manuscript:
>
> * **Clarified Layer Preference:** We have revised **Sec. 3.5 and 4.1** to introduce the **"mode-switching" hypothesis**. This explains that LVP acts as a perturbation to "knock" the model from its default bias (whether that is foreground or background) to the complementary stable mode.
> * **Emergent Properties Literature:** As suggested, we added a discussion on emergent properties in the **Introduction (Sec. 1)** and a detailed review in **Appendix E.3**, situating our work within the context of zero-shot reasoning and correspondence.
> * **Practical Heuristics:** We clarified the practitioner's heuristic: one should first identify the model's default bias using RGB input, then use LVP to probe the alternative hypothesis.
> * **Combining Layers:** We added a discussion to **"Open questions" (Sec. 5)** regarding the blending/combining of multiple hypotheses as a key future direction.
>
> We hope these revisions adequately address your concerns.
>
> We look forward to actively participating in the Author-Reviewer Discussion session and welcome any additional feedback you might have on the manuscript or the changes we have made.
>
> Once again, we sincerely thank you for your contributions to improving the quality of this work.
>
> Best regards,
>
> The Authors of Submission 1835

---

> ### Author Response · Authors · 2025-11-27
>
> Dear Reviewer,
>
> As the discussion phase is ending, we wanted to ensure that our previous response and revisions have adequately answered your questions.
>
> Your insights have helped us significantly improve the paper. If you are satisfied with the changes, we would appreciate it if you would consider updating your rating. Please let us know if there are any final points you'd like us to address or discuss. Thank you!
>
> Best regards,
> Authors of Paper 1835

---

### Official Review · Reviewer_8tiM · 2025-11-01

**Soundness:** 3
**Presentation:** 3
**Contribution:** 3
**Rating:** 6
**Confidence:** 4

**Summary:**

This paper proposes a method can predict the scene's multi-layer structure, where the ambiguity is induced by the transparent surfaces. The proposed Laplacian Visual Prompting (LVP) predicts structure behind the transparent surface, where the Laplacian of the image indicates the high-frequency structure (i.e., the structure behind the often homogeneous surface). To evaluate this, the authors introduced MD-3k, which is a benchmark of the scenes with the ambiguity induced by transparent surfaces. The authors demonstrated the LVP on various MDE models.

**Strengths:**

This paper addresses a novel problem setting with significance in monocular depth estimation.

The proposed Laplacian Visual Prompting is intuitive and effective.

The application of the LVP (especially conditioning the generated geometry) is interesting.

**Weaknesses:**

The assumption of the scene with the transparent surface is a bit strong. In detail, the transparent surface should be fully transparent and smooth, and the structure of the scene behind the surface needs to be sufficiently high-frequency.

**Questions:**

Would the proposed method be generalized to the scenes violating the assumption? For example,
(1) The scene with a smooth glass surface, and the scene behind the surface has only a homogeneous wall.
(2) A glass surface with the patterns (a mosaic or an embossment) and the scene behind the glass surface only includes a smooth, homogenous wall.

If not, what could be the possible way to address this?

Also, the benchmark only includes the scenes following the overall assumption of this paper?

---

> ### Author Response · Authors · 2025-11-13
> **Reply to Reviewer 8tiM**
>
> Dear Reviewer 8tiM,
>
> Thank you for your appreciation of our work. We are delighted to hear that you find our problem setting "novel" and "significant", and our LVP method "intuitive and effective".
>
> We provide the following feedback and clarifications, which are reflected in our revised paper, to address your concerns:
>
>
> ---
>
> >**W.1. Regarding the assumption of the scene (smooth foreground, high-frequency background)**
>
> We agree that the "ideal" scenario (smooth, transparent foreground + high-frequency background) represents a specific assumption. However, we believe it is a reasonable and highly relevant one, as it covers many of the most common ambiguous elements in 3D scenes, such as glass doors and windows.
>
> More importantly, our results are on the **MD-3k benchmark**, which is sampled from diverse, real-world scenes (as discussed in **Sec. 3.2**) and is *not* filtered to only include these ideal cases. The fact that LVP demonstrates a strong ability to decouple layers on this robust benchmark supports our central claim that latent multi-layer knowledge exists and can be generalized. The goal of this work is not to present a perfect, SOTA solution for all forms of transparency, but to use this common case to *demonstrate* a fundamental, hidden capability of these models.
>
>
> ---
>
> > **Q.1. Generalization to scenes violating the assumption (homogeneous background / patterned foreground)**
>
> These are excellent and challenging scenarios. We have now explicitly addressed these failure cases in our new **"Open questions" paragraph (Sec. 5)** and illustrated them in **Fig. 11**.
>
> * **(1) Homogeneous wall (smooth background):** You are correct. This is a clear failure case. LVP relies on amplifying high-frequency signals from the background (as an OOD prompt, see **Sec 3.5**). If the background is a homogeneous wall, its Laplacian signal will be near-zero. In this case, LVP would likely fail to produce a coherent complementary hypothesis.
> * **(2) Patterned glass (high-frequency foreground):** This is a great point. As we now show in **Fig. 11b** (captioned as "semi-transparent surfaces"), our method struggles with patterned or frosted glass. We hypothesize this is because the high-frequency pattern on the *foreground* corrupts the background signal that LVP aims to isolate. Overcoming this is a key direction for future work.
>
>
> ---
> > **Q.2. Does the benchmark only include scenes following the assumption?**
>
> Thank you for this question, as it touches on the core of our paper. The MD-3k benchmark was indeed curated to focus on this common two-layer ambiguity (transparent surface + background) to enable a clean analysis (as described in **Sec. 3.2**).
>
> Our main insight, however, is not to provide a perfect, practical solution for *only* this case. Rather, our goal is to use this case as a clear-cut **demonstration** that these powerful foundation models, despite their single-depth training, possess a latent, multi-layer understanding. We hope this discovery (as stated in our **Sec. 1**) inspires the community to move beyond evaluating models just on their single-task, single-output training objective and to start probing for the richer, emergent capabilities hidden within them.
>
> ---
>
> We would like to thank you again for your insightful and careful suggestions. All of our revisions in the paper are **highlighted in blue** for your convenience. We are **happy to receive further comments and address any remaining concerns**.
>
> If you feel our response and revisions improve the quality of our work, **we would be grateful if you would consider your rating**! Thank you!

---

> ### Author Response · Authors · 2025-11-24
> **Looking Forward to Discussion with You**
>
> Dear Reviewer 8tiM,
>
> We sincerely appreciate your time and effort in reviewing our manuscript and providing valuable feedback.
>
> In response to your insightful comments regarding scene assumptions and generalization, we have made the following revisions to our manuscript:
>
> * **Addressed Scene Assumptions:** We have added a detailed discussion in **"Open questions" (Sec. 5)** and a new **Fig. 11** to explicitly address the failure cases you raised, including homogeneous backgrounds (walls) and patterned/semi-transparent foregrounds.
> * **Clarified Benchmark Scope:** We clarified that while MD-3k focuses on common two-layer ambiguities (transparent surfaces) to allow clean analysis, the method demonstrates a latent capability that exists within the models regardless of the specific scene type.
>
> We hope these revisions adequately address your concerns.
>
> We look forward to actively participating in the Author-Reviewer Discussion session and welcome any additional feedback you might have on the manuscript or the changes we have made.
>
> Once again, we sincerely thank you for your contributions to improving the quality of this work.
>
> Best regards,
>
> The Authors of Submission 1835

---

> ### Author Response · Authors · 2025-11-27
>
> Dear Reviewer,
>
> As the discussion phase is ending, we wanted to ensure that our previous response and revisions have adequately answered your questions.
>
> Your insights have helped us significantly improve the paper. If you are satisfied with the changes, we would appreciate it if you would consider updating your rating. Please let us know if there are any final points you'd like us to address or discuss. Thank you!
>
> Best regards,
> Authors of Paper 1835

---

### Author Response · Authors · 2025-11-13
**Summary of Changes for Our Manuscript Revision**

**Dear Reviewers, Area Chairs, and Program Chairs**,

We sincerely thank the reviewers, ACs, and PCs for the time and effort devoted during this review. We especially appreciate our reviewers for offering valuable comments, providing positive feedback, and drawing insightful suggestions!

---

We are encouraged that our reviewers **find our work and findings to be novel, elegant, meaningful, and interesting**!
* **Reviewer 8tiM:** finds our paper addresses a "novel problem setting with significance," notes the "Laplacian Visual Prompting is intuitive and effective," and that the "application... is interesting."
* **Reviewer sERC:** recognizes that our work "Reveals a previously unreported property," finds the "idea... is elegant and novel," and highlights that it "Provides MD-3k, the dedicated benchmark."
* **Reviewer m2ax:** states the "problem... is challenging, interesting and of practical use," the "core idea... is clever and well-motivated," and the "compilation of a new benchmark is an important contribution."
* **Reviewer YVLM:** notes that our paper "explores... a challenging and meaningful" problem, "introduces MD-3k, a new benchmark," and finds LVP "is simple and proven effective."

---

As suggested by reviewers, we have revised the manuscript accordingly. All revisions in the paper are **highlighted in blue** for better view.

---


We present a **summary of changes** as follows:

### 1. Experimental Analysis & Baselines:
* As suggested by **Reviewer m2ax**, we have added a **"dummy" baseline**  to **Table 2**. This new ideal baseline (56.4% ML-SRA) provides a much stronger reference and highlights our method's **+19.1%** absolute improvement.
* We have clarified the "semantic" method for performance reference in **Sec. 4.2** (as suggested by **Reviewer m2ax**), correcting the bolding and emphasizing that our single-model LVP achieves comparable performance *without* the multiple extra models and geometric assumptions on planar surfaces.
* As requested by **Reviewer m2ax**, we have added the "Same" subset results for the Gaussian (low-pass) prompt to **Table 5**, confirming that high-pass (not low-pass) signals are key to switching modes in ambiguous *Reverse* cases.
* To address **Reviewer YVLM**'s question, we have clarified our Canny edge comparison in **Sec. 4.5**, highlighting that LVP's non-binary, intensity-rich signal provides more robust information than simple binary edges (Table 4).
* To answer **Reviewer YVLM**'s question on curved surfaces, we have added **Fig. 12a**, which explicitly demonstrates LVP's successful generalization to this case.

### 2.  Scope & Future Work:
* As suggested by **Reviewer YVLM**, we have revised the text throughout the paper (e.g., **Sec. 1, 3.1, 5**) to be more precise, referring to our method as "two-layer" where appropriate.
* In response to **Reviewers 8tiM and YVLM**, we have added a new **Fig. 12** and a detailed discussion in our **"Open questions" (Sec. 5)** to explicitly address failure cases, including homogeneous backgrounds and patterned/semi-transparent foregrounds.
* As suggested by **Reviewers m2ax and YVLM**, we list the ordinal-only evaluation as a limitation (due to the lack of real-world metric-scale multi-depth data, a well-known unsolved data and sensor problem) and have cited all the suggested SOTA metric-depth models (UniDepth, Metric3D, etc.) to our paper.



### 3. Elaboration & Writing:
* As suggested by **Reviewers sERC, m2ax, and YVLM**, we have clarified our statement of our core hypothesis. We have replaced the "high-pass = background" intuition with a more robust **"mode-switching" hypothesis** in **Sec. 3.5**.  And we have clarified that this hypothesis only holds for some of existing depth foundation models but not all.
* As suggested by **Reviewer sERC**, we have discussed the literature of  emergent properties of visual models in our **Introduction (Sec. 1)** and included a more detailed review of related works on emerging properties in the appendix, framing our work within this important context.
* We have added a clarification to **Sec. 2** (per **Reviewer YVLM**) distinguishing our deterministic LVP from the non-deterministic, slow sampling of diffusion models.
* To address **Reviewer sERC**'s practical consideration, we have added a discussion on blending/combining hypotheses to our **"Open questions" (Sec. 5)** as a key future direction.

---

For detailed responses regarding each of the above aspects, please kindly refer to the individual rebuttal windows in the review section.

Last but not least, we sincerely thank our reviewers, ACs, and PCs again for the effort devoted and the constructive suggestions provided. **We welcome further discussion and are happy to address any remaining concerns**!

---

### Note · Program_Chairs · 2026-01-17
**Submission Desk Rejected by Program Chairs**

The following references in this submission do not refer to real documents and/or have major errors in bibliographic information:

 Jinsung Bae, Gyeonghyeon Kim, Jihyung Park, Guee-Sang Kim, and Kuk-Jin Yoon. Learning uncertainty-aware monocular depth estimation from unsupervised struct-SDF. In ICCV, 2023.